# Domain Generalizable Person Re-identification via Adversarial Dual-Stream Strategy with Local Consistency

## Abstract

Domain Generalizable Person Re-identification (DG Re-ID) faces significant challenges due to appearance variations across different environments, resulting in domain shifts when models are deployed on unseen target domains. Current methods often neglect shared structural commonalities across identities, which limits their ability to generalize and recognize fine-grained identity details effectively. To address these issues, we propose an Adversarial Dual-Stream Learning (ADSL) framework, which integrates two complementary strategies: mining stable local commonalities and modeling local perturbations. The Cross-Identity Local Consistency Learning (CILL) module builds a memory bank of local features and utilizes clustering-driven similarity learning to balance structural consistency and discriminative granularity. Simultaneously, the Dual-stream Adversarial Perturbation Strategy (DAPS) generates adversarial samples that simulate cross-domain appearance variations while preserving local semantic structures. To further improve robustness to domain shifts, we introduce a Clean-Adv Local Cosine Alignment constraint, which ensures feature consistency between clean and adversarial samples in the local semantic space. Extensive experiments on DG Re-ID benchmarks demonstrate that our method significantly outperforms existing state-of-the-art approaches, highlighting its effectiveness and superiority. The code is available at: https://github.com/STUDY1231/ADSL.

## 1 Introduction

Person re-identification (Re-ID) is a fundamental task in computer vision, aimed at recognizing the same individual across non-overlapping cameras. With the development of deep convolutional neural networks (CNNs) and visual Transformer (ViT) architectures, Re-ID has made remarkable progress within a single domain Li et al. (2019)Tan et al. (2021a)Tan et al. (2022)Xiahou et al. (2023)Tan et al. (2021b)Liu et al. (2020). However, when directly applied to unseen target domains, model performance often experiences a significant decline. This degradation is mainly due to inter-domain distribution discrepancies, which encompass variations in lighting conditions, color shifts, background complexity, camera resolution, and occlusions.

Recent studies have focused on enhancing cross-domain generalization capabilities through global feature modeling and regularization techniques. However, global features are often vulnerable to background noise and local disturbances, hindering the model's ability to capture fine-grained identity differences effectively. In contrast, local features (e.g., head, torso, and leg regions) are more closely aligned with the inherent structural information of pedestrians, providing greater stability and strong identity relevance. These local features play a crucial role in improving cross-domain generalization. Building on this insight, methods such as Ni et al. (2023) Sun et al. (2018) Wu et al. (2023) Guo et al. (2025) have made notable strides by segmenting feature maps and emphasizing locally granular, human-specific discriminative features. However, these approaches fail to explicitly explore cross-identity commonalities in local features, leaving the feature representations vulnerable to domain-specific interference. Besides, the discriminative power of local features is often insufficient, leading to performance degradation when faced with substantial inter-domain variations.

Figure 1: The motivation behind the proposed Adversarial Dual-Stream Learning (ADSL) framework is to address the challenge of large domain shifts. The framework leverages two key strategies: (1) The introduction of the Cross-Identity Local Consistency Learning (CILL) module, which promotes fine-grained discriminative feature learning by learning shared local information across different identities;(2) The Dual-stream Adversarial Perturbation Strategy (DAPS), which generates adversarial perturbation images and aligns the clean and adversarial samples using cosine similarity loss, thereby enhancing the model's robustness to local region features.

To address these challenges, we propose a novel framework for domain generalization in local feature learning, called Adversarial Dual-Stream Learning (ADSL). The motivation of the proposed ADSL is illustrated in Fig. 1. The ADSL framework addresses two major challenges in current methods: (1) the limitations of local feature learning through the introduction of the Cross-Identity Locality Learning (CILL) module, which creates similarity relationships based on visual features rather than identity labels, thus reducing domain bias; and (2) the issue of overfitting to domain-specific features, such as lighting or color variations, by incorporating adversarial perturbations. By introducing adversarial samples during training, we simulate and counteract domain-specific disturbances, enhancing the model's robustness and generalization ability. Clean and adversarial samples are processed simultaneously in a dual-stream fashion, and a cosine similarity loss is employed to align local features between these samples. This encourages the network to focus on shape and structural information that is less sensitive to perturbations, ultimately improving domain generalization and robustness. In summary, our main contributions include:

1. We propose the CILL module, which actively extracts shared local structural features and enhances local discriminability by incorporating a Memory Bank mechanism. It establishes similarity relationships between cross-identity samples within the local feature space, fostering improved local feature representations.

2. We present a novel dual-stream training strategy DAPS that leverages local adversarial perturbations. DAPS explicitly generates and incorporates adversarial samples, significantly enhancing the model's robustness and generalizability against cross-domain visual interference. The strategy is further optimized through Clean-Adv Cosine Feature Alignment and comprehensive optimization of the CILL module, ensuring more reliable performance.

3. We achieve significantly improved performance over existing methods across multiple standard cross-domain Re-ID datasets, demonstrating the effectiveness and strong generalization capability of our proposed approach.

## 2 Related Work

### 2.1 Domain Generalization Person Re-ID

Domain Generalization person Re-Identification (DG Re-ID) primarily aims to enable models trained on a source domain to demonstrate strong generalization capabilities on un-

seen target domains. Existing DG Re-ID approaches can be categorized into several types: domain-invariant feature learningJia et al. (2019a)Zhang et al. (2022a)Luo et al. (2019)Jia et al. (2019b), meta-learning optimizationChoi et al. (2021)Ni et al. (2022)Gong et al. (2023)Du et al. (2024), and data augmentation. The core objective of these approaches is to enhance a model's adaptability across different data domains by learning domain-invariant features or simulating the distribution of the target domain. Domain-invariant feature learning methods aim to eliminate stylistic differences between source and target domains, thereby extracting shared features independent of specific domains. For instance, Jiao et al. (2022)proposed the Dynamic Transformation Instance Normalization (DTIN) method, which reduces inter-domain variations by flexibly adjusting normalization operations. Similarly, Zhao et al. (2021)proposed Memory-based Multi-source Meta-learning (M3L), enabling models to adapt to diverse domains through training within a meta-learning framework. However, these approaches often overlook the loss of discriminative information during normalization, particularly concerning the preservation of identity-related features. This deficiency hinders models from capturing fine-grained identity differences across domains. On the other hand, meta-learning approaches enhance model generalization by simulating domain shifts. Li et al. Li et al. (2018) proposed Meta-Learning Domain Generalization (MLDG), which simulates domain shifts through meta-training and meta-testing sets during training and evaluation, further improving cross-domain performance. While theoretically effective, these methods rely on small distribution differences between source domains; when domain variations are substantial, model generalization may be constrained.

Furthermore, data augmentation methods Zhang et al. (2023)Li et al. (2022)Yang et al. (2024)Zhang et al. (2017)Zhou et al. (2021)Nuriel et al. (2021)Kim et al. (2023)have been extensively applied in DG Re-ID tasks. By perturbing images to expand the diversity of training datasets, these techniques enhance model generalization capabilities. For instance, the uncertainty-based domain transfer (DSU) approach proposed by Li et al. (2022) enhances domain adaptability by utilizing synthetic feature statistics during training. Existing methods in domain generalizable re-identification (DG Re-ID) focus mainly on global feature extraction and style decoding, but still fall short in effectively extracting locally discriminative features that are robust to domain variations. To address this, we propose the Adversarial Dual-Stream Learning (ADSL) framework, which combines local structure modeling with adversarial perturbations to enhance the model's generalization in complex cross-domain scenarios.

## 2.2 Adversarial Learning in Re-ID

Adversarial learning has become a key approach in person Re-ID, focusing on domain-invariant feature extraction and adversarial perturbation generation. For domain-invariant feature extraction, adversarial training reduces feature distribution discrepancies between source and target domains. Early work by Lin et al. (2020) employed adversarial autoencoders to align domain distributions using maximum mean squared distance (MMSD), but this can lead to the loss of discriminative information. To address this, Chen et al. (2021) proposed a dual-constraint method combining domain-aware adversarial learning with identity-aware similarity enhancement, while Zhang et al. (2022b) introduced causal adversarial learning to separate identity and domain-specific factors. Yang & Tian (2022) further analyzed category information loss and used causal inference to address this issue. To enhance adversarial training efficiency, Wei et al. (2024) proposed a dynamic attack strategy, while the Generalizable Metric Network (GMN) Qi et al. (2024) addressed domain shifts using a Metric Network (M-Net) and Dropout-based Perturbation (DP). Liu et al. (2024) improved robustness with the Efficient and Generalized Adversarial Training (EG-AT) method, integrating Universal Adversarial Perturbations (UAP) and Free-AT.

Adversarial perturbation generation methods, such as those proposed by Tan et al. (2025), improve model robustness by synthesizing challenging matching samples across domains. Bian et al. (2025) introduced the Meta-Transfer Generative Attack (MTGA) method to optimize transferable adversarial samples, enhancing resilience across tasks and datasets. For single-source domain-generalized Re-ID, Chen et al. (2025) introduced Multi-Level Feature Perturbation (MLFP) to reduce source-domain bias with Random Background Perturbation

(RBP) and Uncertainty Sampling Normalization Modules (USNM), improving performance on unseen domains.

Building on these advancements, we propose a dual-stream training strategy with local adversarial perturbations, enhancing robustness to real-world disturbances like lighting and color shifts. This improves model generalization and stability in cross-domain Re-ID tasks.

## 3 Method

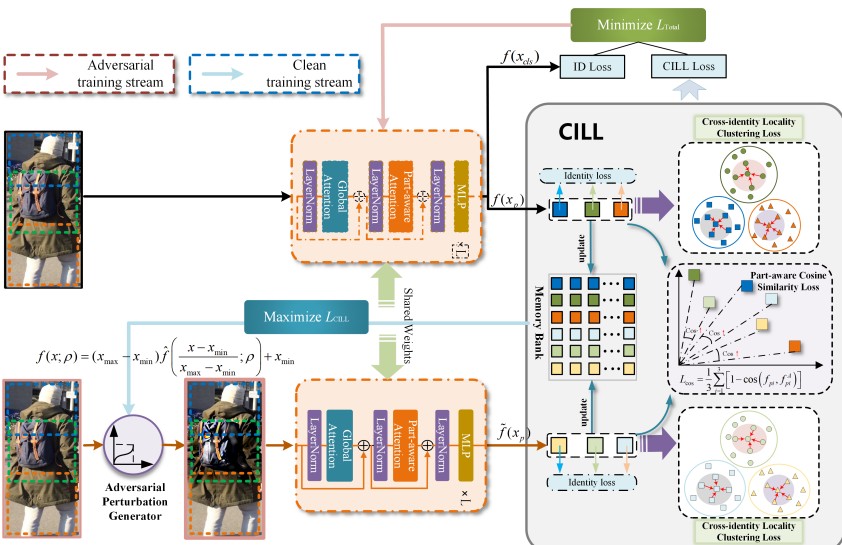

Figure 2: Illustration of the proposed Adversarial Dual-Stream Learning (ADSL) framework that constructs two streams: a clean stream and an adversarial stream, by applying local adversarial perturbations to the input images. Both streams learn discriminative and robust representations through a shared Transformer-based feature extractor. The Cross-Identity Locality Learning (CILL) module is introduced to constrain and cluster the local features of both streams. By jointly optimizing the ID Loss and CILL Loss, the framework enhances cross-domain generalization capability.

### 3.1 Cross-Identity Locality Learning

Local features (such as head, torso, and leg regions) align more closely with the intrinsic structure of pedestrians than global features, providing robust information that is more resistant to domain interference. However, existing methods fail to fully leverage cross-identity shared local features and overlook the importance of discriminative capability, resulting in limitations in fine-grained identity recognition.

To overcome these limitations, we propose the Cross-Identity Locality Learning (CILL) module, which simultaneously optimizes the shared and discriminative properties of local features. By introducing a clustering loss, CILL extracts stable shared local structural features across different identities (as shown in Figure 2), thereby enhancing domain invariance and improving cross-domain generalization performance. During training, each component label $x_{pi}$ maintains a momentum-updated memory repository $\{m_{p_i} \mid i = 1, \ldots, N\}$, whose memory update rule is defined as:

$$m_{p_i}^j = \begin{cases} f\left(x_{p_i}^j\right) & e = 0, \\ (1-\mu) \times m_{p_i}^j + \mu \times f\left(x_{p_i}^j\right) & e > 0 \end{cases} \tag{1}$$

where $e$ denotes the current training epoch; $j = 1, \ldots, K$, where $K$ is the number of samples in the source dataset. $\mu$ is the momentum. For each local feature $f(x_{p_i})$ in the current mini-batch, we compare it against the entire memory bank.

To enable the model to learn cross-identity shared local features, we design a clustering-based loss function utilizing a memory bank. We select the $k$ nearest local features $f(x_{p_i}^j)$ from $\{m_{p_i}\}^K$ to form a positive sample set $\{P_{p_i}^j\}_{j=1}^k$. The loss is defined as:

$$\mathcal{L}_{pi}^j = -\log \frac{\sum\limits_{m_{p_i}^s \in \{P_{p_i}^j\}_{j=1}^k} \exp\left(\frac{f(x_{p_i}^j)m_{p_i}^s}{\tau}\right)}{\sum\limits_{n=1}^N \exp\left(\frac{f(x_{p_i}^j)m_{p_i}^t}{\tau}\right)}, \tag{2}$$

where $\tau$ is the temperature coefficient. By minimizing $\mathcal{L}_{pi}$, the model brings visually similar local regions closer together (sharing structural features across identities) while pushing dissimilar regions farther apart. Additionally, to further enhance the fine-grained discriminative capability of local features, we introduce triplet loss $\mathcal{L}_p^{triplet}$ and cross-entropy loss $\mathcal{L}_p^{ce}$ for each local region. The final optimization objective for CILL is formulated as:

$$\mathcal{L}_{\text{CILL}} = \mathcal{L}_p^j + \sum_{p=1}^Q \left(\mathcal{L}_p^{triplet} + \mathcal{L}_p^{ce}\right). \tag{3}$$

where $Q$ denotes the number of local regions, and $Q = 3$.

### 3.2 Dual-stream Adversarial Perturbation Strategy

Relying solely on clean samples may lead to overfitting to domain-specific attributes, thereby limiting generalization capabilities. Therefore, we propose a dual-stream training strategy based on the fusion of local adversarial perturbations and CILL. Specifically, we design an intensity mapping function $x = f(x; \rho)$ (with trainable parameter $\rho$), which simulates intensity distributions without altering the content of $x$. This adversarial training enriches input diversity, reduces model dependence on specific intensity distributions, and preserves local feature structures to prevent content corruption. The intensity mapping function $\rho$ incorporates $n + 1$ trainable parameters, with intensity adjustments computed via linear interpolation as shown in Equation (4):

$$\hat{f}\left(\tfrac{i}{n}; \rho\right) = \frac{\sum\limits_{j=0}^i e^{\rho_j - \rho_0} - 1}{\sum\limits_{j=0}^n e^{\rho_j - \rho_0} - 1} \in [0, 1], \tag{4}$$

This strategy allows the model to better adapt to domain-specific variations without sacrificing fine-grained discriminative capabilities. Based on this function, we can adjust the intensity of each pixel in the image $x$ according to the formula (5):

$$f(x; \rho) = (x_{\max} - x_{\min})\hat{f}\left(\frac{x - x_{\min}}{x_{\max} - x_{\min}}; \rho\right) + x_{\min} \tag{5}$$

where $\rho = [\rho_0, \cdots, \rho_n]$ denotes the learnable parameters of the attacker, and $f$ guarantees monotonic mapping of intensities with its range restricted to the interval $[x_{\min}, x_{\max}]$.

We introduce masks $M_p$ based on the CILL module's local feature segmentation during adversarial training. Specifically, ViT divides the input image into multiple patches, with each local label corresponding to a set of patches. We leverage this correspondence to define three categories of local features (e.g., head, torso, legs) and assign them corresponding binary masks $M_{p_1}, M_{p_2}, M_{p_3}$. At each adversarial training step, we randomly select a local

region $p \in 1, 2, 3$ and apply the corresponding mask $M_p$. Adversarial samples are generated by $x^A = M_p \cdot f(x, \rho) + (1 - M_p) \cdot x$, where only the pixels in the local region $M_p$ are perturbed. The attacker parameter $\rho$ is updated by maximizing the loss of the CILL module:

$$\hat{\rho} = \arg \max_{\|\rho\|_2 < \delta} L_{\text{CILL}}\big(S(x^A; \theta)\big) \approx \frac{\partial L_{\text{CILL}}}{\partial \rho} \cdot \frac{\delta}{\left\|\frac{\partial L_{\text{CILL}}}{\partial \rho}\right\|} \tag{6}$$

where $S(\cdot)$ denotes the forward pass of the network, $\delta$ controls the perturbation budget, and $\theta$ represents the model parameters. After generating the adversarial sample, we construct a dual-stream training structure using both Clean and Adversarial (Adv) samples. Specifically, for a Clean sample $x$ and its corresponding adversarial counterpart $x^A$, we pass them through the same network $S(\cdot)$ to extract three groups of local features $f_{p1}, f_{p2}, f_{p3}, \tilde{f}p1, \tilde{f}p2, \tilde{f}p3$, as well as the global feature $fcls$ from the Clean stream. We then enforce cosine similarity loss between the local features of Clean and Adv samples:

$$L_{\cos} = \frac{1}{3} \sum_{i=1}^{3} \Big[ 1 - \cos\big(f_{p_i}, \tilde{f}_{p_i}\big) \Big] \tag{7}$$

where $\cos(\cdot)$ denotes cosine similarity. By constraining the directional consistency between clean and adversarial sample features, the model suppresses reliance on unstable low-level features (e.g., brightness, color) and prioritizes structural features (e.g., shape and texture) that remain unchanged under intensity variations. This enables the model to maintain stable representations of local features even when subjected to local perturbations.

### 3.3 Total Loss Function

In addition to the clustering loss and cosine similarity loss, as well as the triplet and cross-entropy losses on local regions used in Sections 3.1 and 3.2, we also incorporate the global triplet and cross-entropy losses of the Clean samples to help the model learn the overall spatial layout and contextual relationships of the entire body. Our overall loss function can be expressed as:

$$L_{\text{total}} = L_{\text{CILL}}^{\text{clean}} + \lambda_1 L_{\text{CILL}}^{\text{adv}} + \lambda_2 L_{\cos} + \big(L_{\text{triplet}}^{\text{clean}} + L_{\text{CE}}^{\text{clean}}\big). \tag{8}$$

Here, $L_{\text{CILL}}^{\text{clean}}$ jointly optimizes clustering and triplet losses on local regions to enhance the cross-identity commonality and fine-grained discriminability of local features; $L_{\text{CILL}}^{\text{adv}}$ strengthens feature consistency under local perturbations; the cosine similarity loss $L_{\cos}$ constrains the alignment of Clean and Adv feature directions, encouraging the model to focus on stable structural characteristics (such as shape and texture) while reducing dependence on unstable factors like illumination and color changes. Finally, the global feature loss $L_{\text{global}}^{\text{clean}} = L_{\text{triplet}}^{\text{clean}} + L_{\text{CE}}^{\text{clean}}$ is constructed from both triplet loss and cross-entropy loss on the Clean stream, providing holistic discriminative capacity based on the entire body, and complementing the optimization of local features.

## 4 Experiments

### 4.1 Dataset and Evaluation Metrics

We evaluate our method on four widely-used public person re-identification (Re-ID) datasets: Market-1501, DukeMTMC-reID, CUHK03, and MSMT17. Market-1501 is a large-scale dataset collected at Tsinghua University using six cameras. It consists of 1,501 identities and 32,668 images, with 751 identities in the training set (12,936 images) and 750 identities in the test set (19,732 images). A face detector was employed to automatically label identities. DukeMTMC-reID is a multi-camera person tracking dataset from Duke University. It includes 702 identities in the training set (16,522 images) and 1,110 identities in the test set (17,661 images), with 408 subjects appearing on a single camera. MSMT17 is a hybrid

indoor-outdoor Re-ID dataset featuring 12 outdoor and 3 indoor cameras. It contains 1,041 identities in the training set (32,621 images) and 3,060 identities in the test set (82,161 images), with complex scene backgrounds and multi-temporal lighting variations. CUHK03 includes surveillance images captured from five camera angles at The Chinese University of Hong Kong. The detection subset, consisting of 767 identities for training and 700 for evaluation, is used in our experiments.

For evaluation, we use Rank-1, which reflects the accuracy of the top-ranked correct match, and mAP (mean Average Precision), which assesses overall retrieval performance by averaging the accuracy across all queries.

Table 1: Comparing data (%) between our method and other domain generalization re-id methods across different source datasets.

| Method | Reference | Source | Duke | | MSMT | | CUHK03 | |
|---|---|---|---|---|---|---|---|---|
| | | | R1 | mAP | R1 | mAP | R1 | mAP |
| SNR | CVPR2020 | | 55.1 | 33.6 | – | – | – | – |
| QAConv | ECCV2020 | | 54.4 | 33.6 | 25.6 | 8.2 | 14.1 | 11.8 |
| TransMatcher | NeurIPS2021 | | – | – | 47.3 | 18.4 | 22.2 | 21.4 |
| QAConv-GS | CVPR2022 | | – | – | 45.9 | 17.2 | 19.1 | 18.1 |
| MDA | CVPR2022 | Market | 56.7 | 34.4 | 33.5 | 11.8 | – | – |
| PAT | ICCV2023 | | 67.9 | 48.9 | 42.8 | 18.2 | 25.4 | 26.0 |
| LDU | IEEE TIM2024 | | 59.5 | 38.0 | 35.7 | 13.5 | 18.5 | 18.2 |
| APD | Neural Net2025 | | 66.4 | 47.6 | – | – | 24.9 | 25.2 |
| PFIEN | KBS 2025 | | 62.3 | 40.1 | – | – | – | – |
| MLFP | IEEE TIM2025 | | 60.8 | 38.8 | 34.3 | 11.9 | – | – |
| DSH | Neural Net2025 | | 58.3 | 37.0 | – | – | – | – |
| ISTDG | Neurocomputing2025 | | 68.7 | 49.5 | – | – | – | – |
| Ours | | | 71.4 | 51.2 | 43.5 | 19.0 | 29.3 | 27.5 |

| Method | Reference | Source | Market | | MSMT | | CUHK03 | |
|---|---|---|---|---|---|---|---|---|
| | | | R1 | mAP | R1 | mAP | R1 | mAP |
| SNR | CVPR2020 | | 66.7 | 33.9 | – | – | – | – |
| QAConv | ECCV2020 | | 62.8 | 31.6 | 32.7 | 10.4 | 11.0 | 9.4 |
| MDA | CVPR2022 | | 70.3 | 38.0 | 39.8 | 13.6 | – | – |
| PAT | ICCV2023 | | 71.9 | 45.2 | 43.9 | 19.8 | 18.8 | 18.9 |
| LDU | IEEE TIM2024 | Duke | 73.2 | 42.2 | 44.2 | 16.7 | 14.2 | 14.2 |
| PFIEN | KBS 2025 | | 73.9 | 42.1 | – | – | – | – |
| APD | Neural Net2025 | | 72.2 | 44.4 | – | – | – | – |
| MLFP | IEEE TIM2025 | | 69.7 | 36.9 | 35.3 | 12.1 | – | – |
| DSH | Neural Net2025 | | 70.4 | 40.4 | – | – | – | – |
| Ours | | | 74.8 | 46.7 | 44.9 | 19.8 | 20.7 | 20.3 |

## 4.2 Implementation Details

We use the ViT-base model with patch size = 16, pretrained on ImageNet, as our backbone (denoted as ViT-B/16). The batch size is set to 64, and the image size is adjusted to $256 \times 128$. To optimize the model, we adopt the SGD optimizer with a weight decay of $10^{-4}$. The learning rate linearly increases from 0 to $10^{-3}$ during the first 10 epochs, and then decays over the following 50 epochs. In total, the training process takes 60 epochs. In the CILL module, we set $\lambda_1$ (weight for the adversarial local loss) and $\lambda_2$ (weight for the cosine similarity loss), as well as the clustering number $k$, to 0.5, 0.2, and 10, respectively. The temperature $\tau$ is set to 0.02, the momentum $\mu$ is set to 0.1, and the attacker budget $\delta$

is set to 3. In addition, the label smoothing parameter is set to 0.1. For the baselines, we follow TransReID-B/16 without SIE and JPM for fair comparison.

### 4.3 Comparison with the SOTA

To verify the performance of the model, we evaluate our framework on single-source generalization ReID benchmark datasets. Specifically, we adopt Market-1501, DukeMTMC-reID, MSMT17, and CUHK03-NP as the training and testing sets. As shown in Table 1, our method achieves Rank-1 improvements of +2.7%, +3.9%, +1.6%, +0.7%, and +1.9% over the current state-of-the-art methods on M→D, M→C03, D→M, D→MS, and D→C03 respectively. For the mAP metric, our method achieves improvements of +2.3%, +0.6%, +1.5%, +1.5%, +0.6%, and +1.4%, on M→D, M→MS, M→C03, D→M, D→MS, and D→C03, respectively. The experimental results demonstrate that our model can more effectively capture cross-identity shared local structural features, validating its superior performance on the person re-identification task.

### 4.4 Ablation Study

To ensure all components contribute to our model, we conducted ablation studies. All models were trained on Market and tested separately on Duke, MSMT, and CUHK 03-NP. We selected the following models: (1) Our baseline model, which introduces local region partitioning on TransReID-B/16 and applies cross-entropy loss and triplet loss to three predefined local regions. (2) The baseline model with the CILL module added. (3) The dual-stream DAPS model with the CILL module and without the cosine similarity loss adversarial perturbation (without Lcos). (4) The dual-stream DAPS model with the CILL module and with the cosine similarity loss adversarial perturbation.

Table 2: Ablation study on the effectiveness of CILL and DAPS.

| CILL | DAPS (without $L_{\cos}$) | DAPS | M→D | | M→MS | | M→C03 | |
|---|---|---|---|---|---|---|---|---|
| | | | R1 | mAP | R1 | mAP | R1 | mAP |
| × | × | × | 66.3 | 47.5 | 39.4 | 16.9 | 24.6 | 25.4 |
| ✓ | × | × | 67.7 | 48.5 | 41.2 | 17.3 | 25.5 | 26.2 |
| ✓ | ✓ | × | 70.3 | 50.7 | 43.0 | 18.5 | 28.5 | 26.9 |
| ✓ | ✓ | ✓ | 71.4 | 51.2 | 43.5 | 19.0 | 29.3 | 27.5 |

Experimental results demonstrate that each component of the proposed model consistently enhances performance across various datasets, validating its effectiveness and generalization capability. Introducing the CILL module improves the baseline model, achieving top accuracies of 67.7%, 41.2%, and 25.5% on the D, MS, and C03-NP settings, respectively. The mean average precision (mAP) also increased to 48.5, 17.3, and 26.2, highlighting that identity-based local feature clustering reduces intra-class variation and improves inter-class separability. Further performance gains were observed with the introduction of the DAPS strategy, incorporating fused adversarial perturbation training. Top accuracies reached 71.4%, 43.5%, and 29.3% under the D, MS, and C03-NP settings, respectively, demonstrating that DAPS enhances robustness by simulating latent variations in image style and structure. These results show that the synergistic interaction between the CILL and DAPS modules significantly boosts overall model performance across multiple metrics.

### 4.5 Parameter analysis

We conducted experiments on key hyperparameters in the CILL module: $\lambda_1$ (adversarial local loss weight) and $\lambda_2$ (cosine similarity alignment loss weight). Figure 3 shows that increasing $\lambda_1$ enhances adversarial features, but too large a value may introduce noise, harming global semantic modeling. The best performance occurs at $\lambda_1 = 0.5$. For $\lambda_2$, increasing it from 0 to 0.2 improves Rank-1 accuracy and mAP, with optimal results at

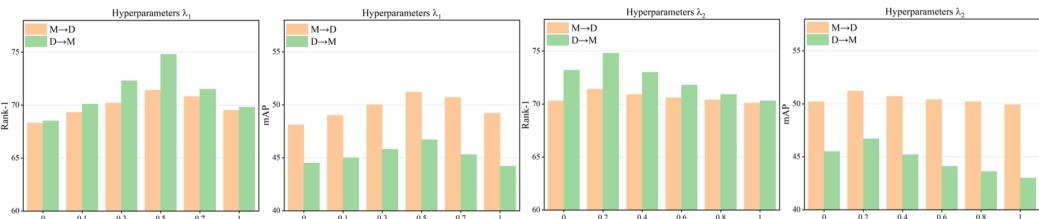

Figure 3: Performance comparison of Rank-1 and mAP for different weight parameters $\lambda_1$ and $\lambda_2$. denotes Market-1501, while 'D' denotes DukeMTMC-reID.

$\lambda_2 = 0.2$. Beyond this, performance declines, suggesting that excessive cosine alignment loss can hinder other objectives and reduce generalization.

### 4.6 T-SNE results

Figure 4 visualizes the feature distributions extracted by our model using t-SNE under the M→D setting. Blue and red represent the source dataset (Market-1501) and target dataset (DukeMTMC-ReID), respectively. Figure 4(a) shows the baseline model, where significant domain shift is evident, with sparse and discrepant feature distributions. In contrast, Figure 4(b) shows the feature distribution from our full framework, where embeddings are more compact and overlap significantly, reducing the domain gap. This improvement is primarily due to the adversarial perturbation module, which enhances robustness to local variations and mitigates inter-domain discrepancies.

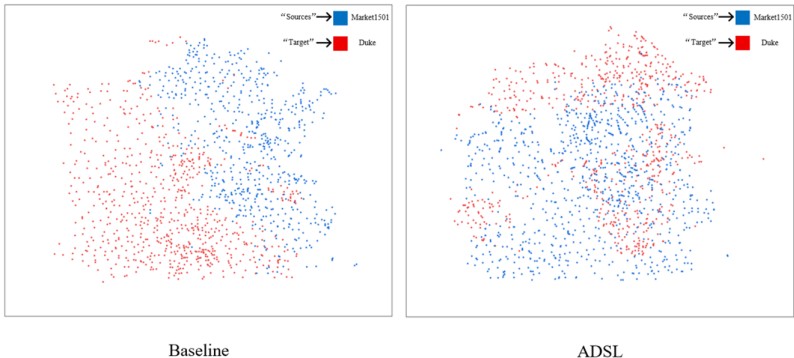

Figure 4: The visualization of feature distributions for the single-source domain M→D on the REID benchmark is provided, generated by the baseline model (without the CILL and DAPS modules) and the full ADSL framework.

## 5 Conclusion

This paper proposes the Adversarial Dual-Stream Learning (ADSL) framework, comprising the Cross-Identity Locality Learning (CILL) module and the Dual-Stream Adversarial Perturbation Strategy (DAPS). The CILL module enhances the model's ability to capture identity-distinctive regions by performing local region clustering across different identities. DAPS strategy employs an adversarial perturbation mechanism to generate clean and adversarial branches. By jointly leveraging these two key parts, it broadens the sample distribution in feature space and enhances consistency, enabling the model to learn disturbance-resistant features. Overall, this approach provides significant insights for advancing domain-general face re-identification technology.

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

# A  Appendix

## A.1  Comparison with the SOTA

As shown in Table 3, our method achieves Rank-1 improvements of +2.8% and +0.8% over the current state-of-the-art methods on C03→ and C03→M respectively. For the mAP metric, our method achieves improvements of +0.1%, +1.3%, and +5.4%,on MS→D, C03→D,and C03→M, respectively. The experimental results demonstrate that our model can more effectively capture cross-identity shared local structural features, validating its superior performance on the person re-identification task.

Table 3: Comparing data (%) between our method and other domain generalization re-id methods across different source datasets.

| Method | Reference | Source | Duke | | Market | | CUHK03 | |
|---|---|---|---|---|---|---|---|---|
| | | | R1 | mAP | R1 | mAP | R1 | mAP |
| SNR | CVPR2020 | | 69.2 | 50.0 | 70.1 | 41.4 | – | – |
| QAConv-GS | CVPR2022 | | – | – | 79.1 | 49.5 | 20.9 | 20.6 |
| TransMatcher | NeurIPS2021 | | – | – | 80.1 | 52.0 | 23.7 | 22.5 |
| QAConv | ECCV2020 | MSMT | 69.4 | 52.6 | 72.6 | 43.1 | 25.3 | 22.6 |
| MDA | CVPR2022 | | 71.7 | 52.6 | 79.5 | 53.0 | – | – |
| LDU | IEEE TIM2024 | | 69.2 | 48.9 | 74.6 | 44.8 | 21.3 | 21.3 |
| APD | Neural Net2025 | | 72.3 | 55.2 | 80.1 | 54.3 | 27.0 | 27.7 |
| Ours | | | 72.0 | 55.3 | 74.5 | 49.0 | 26.1 | 27.2 |
| Method | Reference | Source | Duke | | MSMT | | Market | |
| | | | R1 | mAP | R1 | mAP | R1 | mAP |
| QAConv | ECCV2020 | | 54.0 | 28.5 | 41.8 | 12.6 | 64.2 | 30.2 |
| TransMatcher | NeurIPS2021 | | 57.9 | 35.1 | 44.1 | 14.3 | 70.8 | 39.5 |
| QAConv-GS | CVPR2022 | | 58.5 | 35.6 | 46.4 | 15.7 | 69.1 | 37.4 |
| MTMN | SPL2023 | CUHK03 | 59.3 | 38.3 | 51.8 | 18.1 | 73.0 | 42.0 |
| MLFP | IEEE TIM2025 | | 57.0 | 33.8 | – | – | 70.4 | 39.4 |
| Ours | | | 62.1 | 39.6 | 50.2 | 17.8 | 73.8 | 47.4 |

## A.2  Visualization of Attention Maps

To further verify the capability of our model in local feature modeling, we adopt Grad-CAM to visualize the attention maps corresponding to the global feature (cls token) and the three local features (part1, part2, and part3). As illustrated in Figure 5, the visualization results indicate that the cls token mainly attends to the overall silhouette of the pedestrian, thereby exhibiting strong global discriminative ability. In contrast, the three part tokens focus on different semantic regions of the human body (e.g., head-shoulder, torso, and legs), and their response areas are complementary to each other. This demonstrates that our proposed local modeling mechanism effectively guides the model to perceive semantically meaningful body regions, thereby enhancing both the discriminability and spatial rationality of the learned features. Moreover, even under challenging conditions such as pose variation, occlusion, or clothing changes, the model maintains relatively stable attention to consistent regions. This observation further confirms the positive role of our proposed Dual Adversarial Pathway Strategy (DAPS) in improving the model's robustness against uncertain factors.

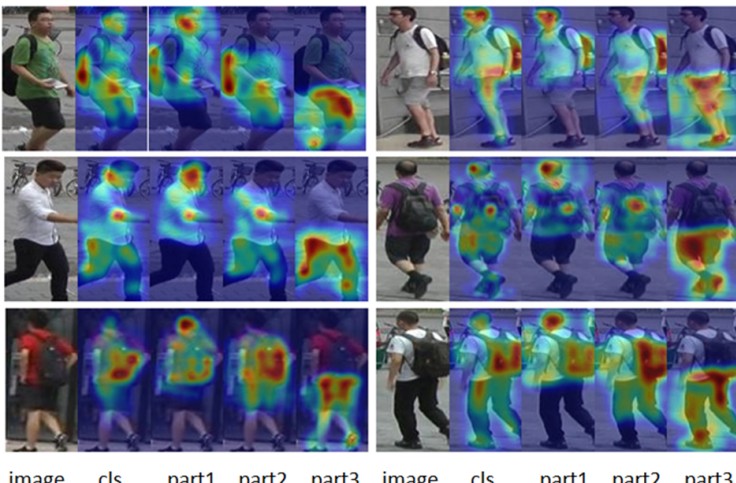

Figure 5: Original images are selected from the target domain (Market-1501). We exhibit the visualizations of the class token and three part tokens.

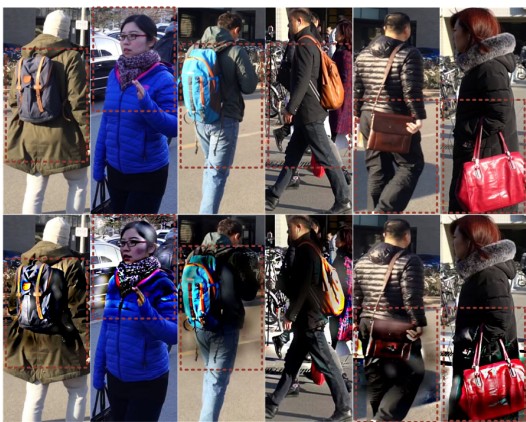

Figure 6: As shown on the MSMT17 dataset, we visualize the effects of the adversarial perturbation mechanism proposed in our method, comparing the sample images before and after perturbation in local regions. The first row of each image group shows the original image (Clean), while the second row displays the image with adversarial perturbations applied to the local regions (Adversarial). The areas marked with red dashed boxes in the figure indicate the regions that were perturbed during the training phase.

## A.3 Visualization of Adversarial Samples

As shown in Figure 6, we further conduct visualization of adversarial samples at the image level on the MSMT17 dataset. Each row displays a comparison between clean images and their corresponding adversarial counterparts. It can be observed that our designed perturbation strategy does not indiscriminately affect the entire image but instead applies perturbations randomly to masked regions. These perturbations manifest as changes in image contrast, brightness, or texture details, which effectively interfere with the model's robustness to local regions in the feature space. This strategy explicitly guides the model to focus on the impact of local variations on discriminative performance, thereby improving its adaptability to style variations across domains. Furthermore, when combined with our proposed Consistency-aware Local Learning (CILL) module, additional constraints are imposed on the cosine similarity between clean and adversarial samples in the local feature space. This effectively enables the model to maintain its original discriminative ability

while enhancing tolerance to local perturbations. In this way, the model achieves more robust perception and utilization of local information, leading to superior cross-domain person re-identification performance.

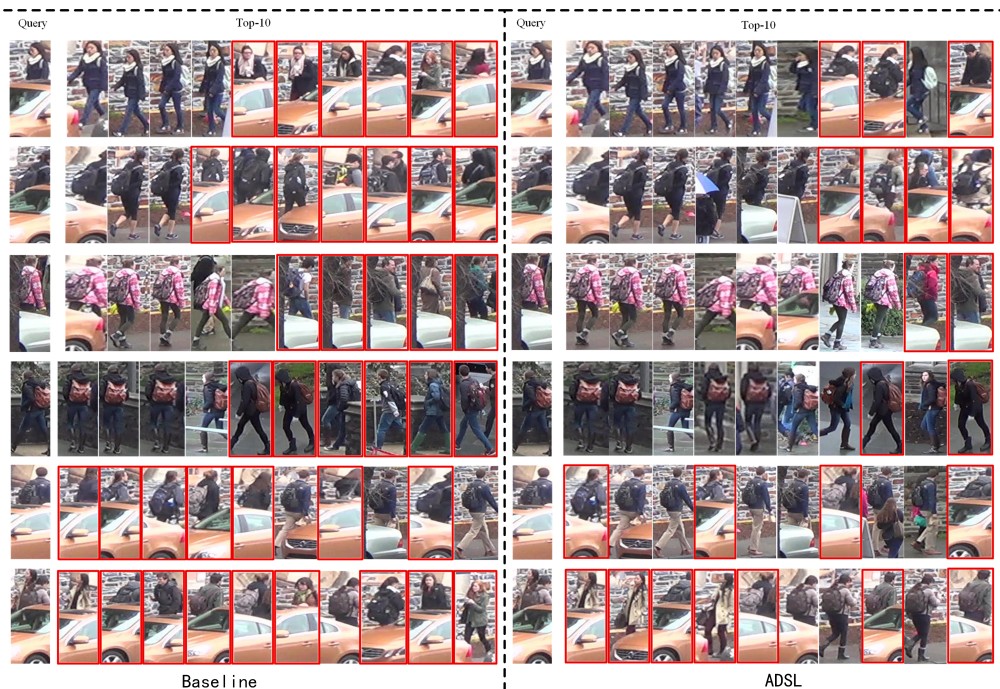

Figure 7: Comparison of retrieval performance between the baseline and ADSL models on severely occluded query images from the Duke dataset. This experiment was conducted under the single-source M→D setting. The top-10 retrieved images for each query are displayed, with red bounding boxes highlighting the mismatched results.

### A.4 Top-10 Retrieval Results

To further validate the robustness of our model in handling occlusions, we present a visual comparison of the retrieval performance between the baseline and ADSL models in the Market→Duke setting, as shown in Figure 7. The images in this figure show query samples from the Duke dataset, where significant occlusions—such as missing legs—present challenges to the model's ability to accurately identify the person.

In the results obtained using the ADSL model, we observe that, even in the presence of occlusions, the model maintains relatively high retrieval accuracy, correctly identifying the target identity in most cases. This is particularly evident in the second row of the figure, where the query image is severely occluded around the legs. Despite the occlusion, ADSL retrieves a more accurate match compared to the baseline. This success is attributed to the Cross-Identity Locality Learning (CILL) module, which focuses on learning shared, stable structural features (such as the torso and head) across identities, thus enabling the model to better generalize to partial or occluded inputs. The CILL module reduces domain bias by establishing similarity relationships based on local visual features rather than identity labels, ensuring that the model can rely on these local features even when key regions are occluded. As a result, ADSL maintains better retrieval performance under occluded conditions than the baseline model, which may overfit to domain-specific attributes like color or background and fail when such attributes are missing.

However, in the case of extreme occlusions, we observe some failure scenarios. In the fifth row of the figure, where the occlusion is much more severe, ADSL fails to retrieve the correct identity for the query image, as indicated by the red bounding box. This case highlights

the challenge that even ADSL faces when large portions of the body, such as the torso and legs, are occluded. While the CILL module helps the model focus on stable features in less occluded regions, it still struggles when too many key features are missing. Despite this, ADSL still demonstrates an overall advantage. For example, in the remaining queries from the same row, ADSL retrieves much better matches compared to the baseline. This suggests that the Dual-Stream Adversarial Perturbation Strategy (DAPS), which simulates domain-specific perturbations and forces the model to focus on structural features, enables ADSL to retain stronger generalization abilities. DAPS helps prevent the model from overfitting to domain-specific features (such as lighting or background), allowing it to handle missing or occluded regions more effectively than the baseline.

These failure cases provide important insights for future work. One area for improvement is the development of more advanced local feature fusion techniques that can better handle the loss of critical body parts, particularly in extreme occlusion scenarios. The combination of CILL and DAPS has already improved robustness by allowing the model to leverage stable features and reduce domain bias. However, integrating additional mechanisms to better handle severe occlusions—such as incorporating temporal or multi-modal information—could further enhance performance in these scenarios. Additionally, fine-tuning the model on occlusion-specific datasets could help improve its ability to generalize to extreme occlusion cases, ensuring better overall performance under such conditions.

A.5   Training Overhead Analysis

Table 4 presents the computational overhead of the ADSL framework compared to the baseline model. While the GFLOPS and parameter size remain consistent across all configurations, the introduction of CILL and DAPS results in increased training time per epoch. Specifically, the baseline model requires 5.61 minutes per epoch for Market→Duke, whereas adding CILL increases training time to 5.72 minutes. The inclusion of DAPS significantly raises training time to 16.2 minutes for Market→Duke and 13.5 minutes for Duke→Market, primarily due to the additional adversarial perturbation generation and memory bank updates.

Despite the increased training time, inference time remains stable across all configurations, highlighting that the additional computational cost of ADSL mainly affects the training phase, with negligible impact on model inference efficiency.

Table 4: Training overhead of ADSL compared to the baseline model.

| Market->Duke | | | | |
|---|---|---|---|---|
| Method | GFLOPS | Params(M) | Training Time (min/epoch) | Inference time (ms/img) |
| baseline | 11.3 | 87.1 | 5.61 | 63.7 |
| baseline+CILL | 11.3 | 87.1 | 5.72 | 63.7 |
| baseline+CILL+DAPS | 11.3 | 87.1 | 16.2 | 63.7 |
| Duke->Market | | | | |
| Method | GFLOPS | Params(M) | Training Time (min/epoch) | Inference time (ms/img) |
| baseline | 11.3 | 87.1 | 4.63 | 30.2 |
| baseline+CILL | 11.3 | 87.1 | 4.74 | 30.2 |
| baseline+CILL+DAPS | 11.3 | 87.1 | 13.5 | 30.2 |

A.6   Training Stability of Adversarial Perturbation Module

Figure 8 illustrates the training trends for loss and accuracy on the Market and Duke datasets. We observe that the inclusion of CILL + DAPS does not introduce training instability. The loss decreases steadily across epochs for all configurations, and the accuracy improves similarly for the baseline, baseline + CILL, and baseline + CILL + DAPS models. This demonstrates that the adversarial perturbation module can be integrated into the training process without causing divergence or instability. The loss curves for CILL and CILL + DAPS are slightly higher during the initial epochs compared to the baseline model,

but they eventually converge to similar levels of accuracy by the end of the training, showing stable learning dynamics.

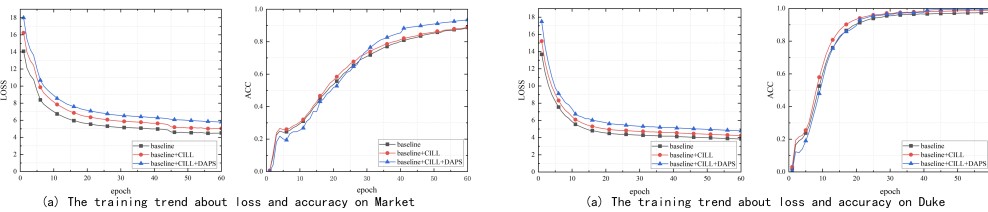

Figure 8: Training trends for loss and accuracy on market and duke datasets.

