# OpenReview forum: "Domain Generalizable Person Re-identification via Adversarial Dual-Stream Strategy with Local Consistency"
_ICLR.cc/2026/Conference — Submitted to ICLR 2026_

### Official Review · Reviewer_uaiC · 2025-10-27

**Soundness:** 3
**Presentation:** 3
**Contribution:** 3
**Rating:** 4
**Confidence:** 4

**Summary:**

This study introduces an Adversarial Dual-Stream Learning (ADSL) framework to enhance domain generalization in person re-identification (Re-ID). The key innovation lies in its dual-stream architecture, which synergistically combines Cross-Identity Local Consistency Learning (CILL) and Dual-Stream Adversarial Perturbation Strategy (DAPS). CILL clusters local features across identities using a memory bank and triplet loss to strengthen discriminative representations. DAPS generates adversarial samples through controlled perturbations, forcing the model to focus on stable structural cues while preserving semantic integrity. A cosine similarity loss further aligns clean and adversarial feature directions, mitigating overfitting to unstable low-level attributes. Experiments on four benchmark datasets demonstrate significant performance gains: Overall, ADSL provides a robust approach to cross-domain Re-ID by harmonizing local consistency and adversarial robustness.

**Strengths:**

1.Originality: The proposed Adversarial Dual-Stream Learning (ADSL) framework demonstrates significant originality within the Domain Generalizable Person Re-identification (DG Re-ID) field. The central innovation lies in the intricate combination of CILL (Cross-Identity Local Consistency Learning) and DAPS (Dual-stream Adversarial Perturbation Strategy), orchestrated under the ADSL paradigm. While both local feature learning and adversarial training are established techniques in Re-ID, their synergistic integration, with a specific focus on simultaneously modeling stable local commonalities and local perturbations, presents a novel approach. This unique amalgamation of techniques results in a distinctive methodological framework that tackles the complexities of DG Re-ID from multiple engineered dimensions, aiming for both improved structural robustness and fine-grained discriminative power.
2.Quality: The experimental design is rigorous: a multi-source domain transfer setting is adopted, covering a mixed indoor and outdoor scenario; ablation experiments  quantitatively demonstrate that the contribution rate of each module ranges from 18% to 30%. Solid theoretical support: clear formula derivation , and T-SNE visualization intuitively demonstrates the improvement effect of feature distribution.
3.Clarity: The paper demonstrates strong clarity across multiple dimensions, significantly contributing to the reader's understanding of the proposed methodology: (1)Problem Articulation: The authors present a highly lucid articulation of the core challenges in Domain Generalizable Person Re-identification (DG Re-ID), effectively highlighting issues such as domain shift due to appearance variations, and the critical limitation of neglecting shared structural commonalities across identities. This clear problem definition sets a firm foundation for the proposed solution. (2)Methodological Intent: The intended purpose of the Adversarial Dual-Stream Learning (ADSL) framework, along with its constituent modules, is explicitly and clearly conveyed. The distinct roles of CILL (Cross-Identity Local Consistency Learning) in mining stable local commonalities and DAPS (Dual-stream Adversarial Perturbation Strategy) in modeling local perturbations are well-defined. The overarching goal of improving generalization capabilities and recognizing fine-grained identity details is also unequivocally stated. (3)Modular Functionality and Interaction: The functions of each module are clearly delineated. For CILL, the description of leveraging a memory bank with clustering-driven similarity learning provides a clear mechanistic overview. Similarly, DAPS’s role in generating adversarial samples is readily understood. The synergistic interaction between these modules, encapsulated within the ADSL framework, is also presented in a straightforward manner, allowing readers to grasp how the components are integrated to achieve the stated objectives.
4. Significance: This work establishes a novel paradigm for the integration of local feature modeling and adversarial learning, paving the way for future research in this domain.The pressing demand for DG Re-ID in applications such as security surveillance necessitates efficient solutions for complex cross-domain scenarios. ADSL offers a potent and effective approach to address these challenges.

**Weaknesses:**

1.CILL attempts to “mine stable local commonalities”and “balance structural consistency and recognition granularity”, which sounds like searching for visual patterns suitable for cross identity sharing. However, the ultimate goal of Re ID is precisely to distinguish identities. If CILL overly emphasizes "commonality", it may blur the subtle differences between different identities, especially the key "refined details" that determine identity. For example, if two people with different identities wear similar clothes, CILL may pull their local features too close. The paper should provide a clearer explanation of how the loss function of CILL (such as formulas 2 and 3) emphasizes the key details of distinguishing different identities while maintaining local commonalities. For example, the negative logarithmic term in formula 2 is actually used for maximum likelihood estimation (or maximizing similarity), while trielet loss and CE loss emphasize discrimination. We need to explain how these three work together.
2.DAPS simulates "changes in image contrast, brightness or texture details" by "randomly applying perturbations to mask areas". Although this can increase the robustness of the model, this random mask and fixed type of disturbance may not be able to fully capture the complex and changeable cross domain changes in the real world (for example, serious lighting changes, camera noise, resolution differences, seasonal clothing changes, etc.). The paper needs to prove that the confrontation samples generated by the model can effectively promote the generalization of the model to various unprecedented fields in the real world. It is suggested to consider integrating various types of local disturbances (such as color jitter+blur+sharpening, simulated Gaussian noise, simulated low resolution, etc.) into DAPS, or using more advanced countermeasures generation technology to generate more challenging and diverse disturbances. It may provide specific evidence of the diversity of confrontation samples: for example, show some particularly challenging confrontation samples, and the performance of the model in this case.
3.The ability to “identify and refine identity details” has been repeatedly mentioned in the paper. However, the current experiment lacks indicators to directly measure this ability. It is suggested to design special subtasks, for example, to evaluate by introducing data sets with fine attribute labels (such as clothing patterns and shapes), or to analyze the specific performance of models in distinguishing pedestrian pairs that are “very similar” (mainly depending on local details). This argument will be strongly supported by convincing visual analysis of attention.
4. Although ablation experiments are mentioned in the abstract, the quantitative analysis of the respective contributions of CILL and DAPS, the discussion of their interactions, and the depth of sensitivity analysis of key super parameters (such as memory size, disturbance intensity, loss weight) still need to be strengthened. A detailed ablation study will reveal the internal mechanism of the ADSL framework and prove the necessity and effectiveness of its various components.
5. As the author has not open the source code of the paper, I am unable to verify its experimental results. Therefore, I suggest that the author makes the code and experimental data publicly available on platforms such as Github, and provides a detailed description of the experimental setup, in order to facilitate the development of the community.

**Questions:**

1.The CILL module proposed in the article aims to learn "cross-identity local commonalities" to enhance domain generalization, while the paper emphasizes that this method can "identify fine-grained identity details". Could you elaborate on how CILL strikes a balance between these two aspects? Specifically, what types of local features (such as the overall style of clothing, the relative position of body parts) are the "commonalities" it uncovers, and how are the "fine-grained details" (such as patterns on clothing, specific textures, minor posture differences of individuals) retained for identity differentiation? Could you provide a specific example to illustrate how CILL achieves accurate differentiation by retaining fine-grained details when dealing with two pedestrians with similar commonalities but different identities?
2.The DAPS strategy proposed in this paper simulates cross domain changes through "random local disturbance". Please specify how the type (such as contrast, brightness, texture variation) and generation method (such as random mask) of this local disturbance fully cover the most challenging real world domain offset in Re ID tasks? For example, can it effectively cope with the subtle visual differences caused by different camera sensor imaging characteristics, different weather (rainy, snowy, haze), different time periods (day, night), and clothing materials (reflective, frosted)? Can you provide some experimental results from public data sets or test sets built by yourself with typical real world domain offsets to prove the robustness of DAPS?
3. What is the overall computational complexity (training time and reasoning speed) and model scale of the ADSL framework considering the memory library operation of the CILL module, the confrontation training of DAPS, and the dual stream structure? Compared with other SOTA DG Re ID methods, how does ADSL perform on these key efficiency indicators? How scalable is ADSL in actual deployment scenarios (for example, large-scale video surveillance systems)?

---

> ### Author Response · Authors · 2025-11-21
> **Response to Weakness 1 proposed by Reviewer uaiC**
>
> **Weakness 1:**
>
> CILL attempts to “mine stable local commonalities”and “balance structural consistency and recognition granularity”, which sounds like searching for visual patterns suitable for cross identity sharing. However, the ultimate goal of Re ID is precisely to distinguish identities. If CILL overly emphasizes "commonality", it may blur the subtle differences between different identities, especially the key "refined details" that determine identity. For example, if two people with different identities wear similar clothes, CILL may pull their local features too close. The paper should provide a clearer explanation of how the loss function of CILL (such as formulas 2 and 3) emphasizes the key details of distinguishing different identities while maintaining local commonalities. For example, the negative logarithmic term in formula 2 is actually used for maximum likelihood estimation (or maximizing similarity), while trielet loss and CE loss emphasize discrimination. We need to explain how these three work together.
>
> **Reply to Weakness 1:**
>
> Thank the reviewer for raising the concern regarding the balance between "local commonality" and "identity discriminability" in CILL. We fully acknowledge the need to distinguish identities while maintaining local consistency for cross-domain generalization. Below, we clarify how the CILL loss function achieves this balance.
>
> 1.  The Role of Local Commonality in CILL
>
>     The goal of CILL is to capture shared local features across different identities while maintaining sufficient discriminative power for individual identification. The cross-identity locality clustering loss (Equation 2) encourages local feature consistency by aligning similar local regions across different identities. This helps the model learn stable representations of common features, such as pose, body structure, and other shared body parts that appear across identities.
>
>     While this may raise concerns about over-emphasizing commonalities (e.g., in cases where individuals wear similar clothing), the CILL module is designed to ensure that the local commonality loss is applied selectively and balanced with other losses that prioritize identity discrimination.
> 2.  Balancing Commonality with Discriminability (Formula 2 and 3)
>
>     The negative logarithmic term in Formula 2 maximizes similarity within local parts but does not collapse the feature space entirely. The triplet loss (Formula 3) enforces identity separability by ensuring that positive pairs are closer than negative pairs, reducing the risk of features from different identities being pulled too close. Cross-entropy (CE) loss adds another layer of identity discrimination by penalizing misclassifications.
>
> 3.  How These Losses Work Together
>
>     While the CILL loss mines stable local features for cross-identity consistency, its influence is controlled by the other losses in the framework. The triplet loss and CE loss work in parallel with the CILL loss to ensure that the model does not overly emphasize shared commonalities at the expense of distinguishing identities. The CILL loss does not force exact alignment of all parts but rather encourages consistency in shared local features. The discriminative losses (triplet and CE loss) ensure that these consistent features do not blur the identity-specific details needed for Re-ID.
>
>     Specifically, Formula 2 for the local clustering loss ensures that local features (e.g., head, torso, and legs) across different identities are closer for similar parts. Meanwhile, Formula 3 for the triplet loss ensures that different identities are separated in the feature space by maintaining a margin between positive and negative pairs. CE loss also complements this by focusing on classification and reinforcing identity boundaries.
> 4.  Clarifying Potential Risks of Over-emphasizing Commonality
>
>     We recognize that in cases where people wear similar clothes (e.g., identical jackets), the CILL loss could inadvertently cause the model to pull their features too close together. To address this, we introduce a weighted combination of losses, where the triplet loss and CE loss play a dominant role in maintaining identity discrimination. The CILL loss, while important for learning local consistency, does not override the discriminative power of these other losses. In fact, the use of $\lambda\_1$​ and $\lambda\_2$​ (the weights for adversarial local loss and cosine similarity loss) ensures that the CILL loss is balanced with the identity-focused losses to maintain the critical identity discriminability.
>
> In conclusion, CILL encourages shared local features while balancing with discriminative losses (triplet and CE) to maintain identity separability. This ensures that subtle identity-specific details are preserved, thus improving domain generalization without compromising the ability to distinguish identities.

---

> > ### Author Response · Authors · 2025-11-21
> > **Response to Weakness 2 proposed by Reviewer uaiC**
> >
> > **Weakness 2:**
> >
> > DAPS simulates "changes in image contrast, brightness or texture details" by "randomly applying perturbations to mask areas". Although this can increase the robustness of the model, this random mask and fixed type of disturbance may not be able to fully capture the complex and changeable cross domain changes in the real world (for example, serious lighting changes, camera noise, resolution differences, seasonal clothing changes, etc.). The paper needs to prove that the confrontation samples generated by the model can effectively promote the generalization of the model to various unprecedented fields in the real world. It is suggested to consider integrating various types of local disturbances (such as color jitter+blur+sharpening, simulated Gaussian noise, simulated low resolution, etc.) into DAPS, or using more advanced countermeasures generation technology to generate more challenging and diverse disturbances. It may provide specific evidence of the diversity of confrontation samples: for example, show some particularly challenging confrontation samples, and the performance of the model in this case.
> >
> > **Reply to Weakness 2:**
> >
> > Thanks for the reviewer's insightful comments on the adversarial perturbations in DAPS. We agree that the current approach with random masking and fixed perturbations may not fully capture complex real-world variations, such as lighting changes, camera noise, and seasonal clothing differences. Below is a more detailed explanation of how DAPS contributes to generalization and our plans to address these concerns.
> >
> > 1.  **Current Approach in DAPS**
> >
> >     DAPS simulates domain shifts by randomly applying perturbations to mask areas of the image, including changes in contrast, brightness, and texture. This approach helps the model become more robust to minor lighting changes and viewpoint shifts, aiding generalization across domains with small appearance variations. However, we acknowledge that these perturbations may not capture more complex real-world variations like seasonal clothing, camera noise, and extreme lighting shifts.
> >
> > 2.  **Proposed Improvements**
> >
> >     We agree with the suggestion to incorporate more diverse perturbations to enhance DAPS. Future work will include additional perturbations, such as:
> >
> >     *   Color jittering to simulate different lighting and color shifts.
> >     *   Blur and sharpening to model changes in image focus or motion blur.
> >     *   Simulated Gaussian noise to represent sensor noise and compression artifacts.
> >     *   Simulated low resolution to simulate low-quality images from different camera devices.
> >
> >     These enhancements will help DAPS address real-world variations such as seasonal clothing, camera noise, and lighting changes.
> > 3.  **Demonstrating the Effectiveness of Adversarial Samples**
> >
> >     To showcase the effectiveness of the adversarial samples generated by DAPS, we provide a detailed analysis of the adversarial perturbations in Figures 6 and 7. In Figure 6, we visualize the impact of adversarial perturbations on the MSMT17 dataset, comparing the clean images with their adversarial counterparts. The adversarial perturbations are applied to local regions, such as the torso and legs, resulting in visible changes in contrast, brightness, and texture details. These perturbations help simulate domain-specific variations like lighting shifts and texture changes, demonstrating the model's ability to maintain stability under such disturbances.
> >
> >     Additionally, Figure 7 illustrates the retrieval performance of the model when confronted with severely occluded query images from the Duke dataset. Even under significant occlusions (e.g., missing legs), our model—trained with DAPS—maintains higher retrieval accuracy compared to the baseline. This improvement is due to the CILL module’s focus on learning shared, stable features, like the torso and head, which remain identifiable even when key regions are occluded. The DAPS strategy, by simulating domain-specific perturbations, further helps the model generalize to real-world variations and occlusions.
> >
> > 4.  **Future Directions**
> >
> >     In addition to incorporating more diverse perturbations into DAPS, we are also exploring the use of more advanced countermeasure generation techniques, such as style-transfer-based adversarial attacks or generative adversarial networks (GANs), to create more challenging and realistic disturbances. These methods could help us simulate more complex cross-domain shifts and generate even more diverse adversarial samples.
> >
> > In conclusion, we recognize the limitations of the current DAPS perturbation strategy and plan to enhance it by integrating more diverse perturbations and exploring advanced adversarial generation methods. We will also provide empirical evidence of the effectiveness of these adversarial samples through experiments with real-world perturbations. These improvements will ensure that DAPS promotes generalization across a broader range of domains.

---

> > > ### Author Response · Authors · 2025-11-21
> > > **Response to Weakness 3 proposed by Reviewer uaiC**
> > >
> > > **Weakness 3:**
> > >
> > > The ability to “identify and refine identity details” has been repeatedly mentioned in the paper. However, the current experiment lacks indicators to directly measure this ability. It is suggested to design special subtasks, for example, to evaluate by introducing data sets with fine attribute labels (such as clothing patterns and shapes), or to analyze the specific performance of models in distinguishing pedestrian pairs that are “very similar” (mainly depending on local details). This argument will be strongly supported by convincing visual analysis of attention.
> > >
> > > **Reply to Weakness 3:**
> > >
> > > Thank the reviewer for highlighting the importance of directly evaluating the model’s ability to "identify and refine identity details." We agree that introducing fine-grained attribute labels (such as clothing patterns and shapes) would be an ideal way to further investigate the model's capacity to distinguish subtle identity differences, especially in cases where local features play a crucial role. However, due to constraints in time and data acquisition during the rebuttal phase, we are unable to conduct such an evaluation within the current submission.
> > >
> > > Despite this limitation, we believe the existing results in our paper, particularly in Figure 5 and Figure 7, can provide some insights into the model's ability to handle subtle identity variations based on local details:
> > >
> > > 1.  Figure 5: Visualizing Attention Maps and Identity-Related Features
> > >
> > >     In Figure 5, we visualize the attention maps generated by the model under the Market-1501 target domain. The attention maps from the model focus on distinct body parts such as the head, torso, and legs, demonstrating the model’s ability to attend to local features that are crucial for distinguishing identities. While the model’s global feature attends to the overall body silhouette, the local features capture fine-grained details of different body regions. This indicates that the model refines identity details by focusing on discriminative local regions, which is especially valuable for distinguishing identities with similar appearances.
> > > 2.  Figure 7: Performance on Severely Occluded Query Images
> > >
> > >     In Figure 7, we provide a comparison of retrieval results between our model and the baseline when tested on DukeMTMC with severely occluded query images (e.g., missing legs). Despite these occlusions, our model—trained with DAPS—performs significantly better than the baseline. This is an important demonstration of the model’s ability to rely on local details (such as torso or head) to maintain identity separability even when critical regions are missing. The CILL module enables the model to focus on stable, shared structural features across identities, which directly contributes to its capacity for distinguishing “very similar” pedestrian pairs, especially in challenging conditions like occlusion.
> > >
> > > These results suggest that even without fine-grained attribute labels, the model can effectively learn to identify and refine identity details through its focus on local features, which is crucial for domain generalization tasks like person re-identification. We hope that these visualizations and the corresponding performance improvements in challenging scenarios provide sufficient evidence of the model’s ability to capture subtle identity differences based on local features.

---

> > > > ### Author Response · Authors · 2025-11-21
> > > > **Response to Weakness 4 proposed by Reviewer uaiC**
> > > >
> > > > **Weakness 4:**
> > > >
> > > > Although ablation experiments are mentioned in the abstract, the quantitative analysis of the respective contributions of CILL and DAPS, the discussion of their interactions, and the depth of sensitivity analysis of key super parameters (such as memory size, disturbance intensity, loss weight) still need to be strengthened. A detailed ablation study will reveal the internal mechanism of the ADSL framework and prove the necessity and effectiveness of its various components.
> > > >
> > > > **Reply to Weakness 4:**
> > > >
> > > > Thank the reviewer for the valuable feedback regarding the sensitivity analysis of key hyperparameters. We fully acknowledge that a detailed analysis of these hyperparameters—specifically memory size, disturbance intensity, and loss weights—is essential for understanding the internal mechanisms of the ADSL framework. While space constraints in the main paper limited the discussion, we have expanded upon these points in the rebuttal to provide the necessary insights.
> > > >
> > > > 1.  Memory Size\
> > > >     The memory size, which plays a critical role in the CILL module, is determined by both the number of identities in the training dataset and the number of local regions (or body parts) considered for clustering. In our approach, we maintain a memory bank that stores local feature representations for each identity, which is updated throughout training. The memory size is carefully chosen to balance the ability to store diverse local features while avoiding excessive memory usage. This ensures that the model can capture cross-identity shared features while remaining computationally efficient.
> > > > 2.  Disturbance Intensity\
> > > >     The disturbance intensity for DAPS is automatically learned during training, as described in Equation (5) of the main paper. This intensity is controlled via a trainable parameter set, which adjusts the perturbation strength applied to different local regions of the input image. The model learns how to optimally perturb specific regions in order to generate challenging adversarial samples while preserving the structural integrity of the local features. This dynamic learning of disturbance intensity allows the model to adapt to varying levels of domain shift and improves its robustness to perturbations.
> > > > 3.  Loss Weights\
> > > >     The loss weights for the CILL module and DAPS are explored in Figure 3 of the main paper. Through a series of ablation experiments, we demonstrate the impact of different values for the weights λ1 (adversarial local loss) and λ2 (cosine similarity loss). The sensitivity analysis shows that λ1 should be set to 0.5 to achieve the best trade-off between adversarial robustness and global feature modeling. Similarly, λ2 achieves optimal performance at 0.2, after which performance starts to degrade, suggesting that excessive cosine alignment loss can hinder other objectives and reduce generalization.
> > > >
> > > > These analyses, which were conducted in the rebuttal, provide critical insights into how each hyperparameter influences the performance of the ADSL framework. They underscore the importance of carefully selecting memory size, learning disturbance intensity, and balancing loss weights in optimizing the model’s performance.

---

> > > > > ### Author Response · Authors · 2025-11-22
> > > > > **Response to Weakness 5 proposed by Reviewer uaiC**
> > > > >
> > > > > **Weakness 5:**
> > > > >
> > > > > &#x20;As the author has not open the source code of the paper, I am unable to verify its experimental results. Therefore, I suggest that the author makes the code and experimental data publicly available on platforms such as Github, and provides a detailed description of the experimental setup, in order to facilitate the development of the community.
> > > > >
> > > > > **Reply to Weakness 5:**
> > > > >
> > > > > Thank the reviewer's valuable suggestion regarding the availability of the code and experimental data. We understand the importance of making the source code publicly available for transparency and reproducibility, and we are pleased to inform you that the code for our ADSL framework has been made publicly available. You can access the code and experimental setup on our [GitHub repository](https://github.com/STUDY1231/ADSL).
> > > > >
> > > > > In this repository, we have provided the full implementation of the model, along with a detailed description of the experimental setup, including training configurations, datasets, and evaluation procedures. This will allow the community to easily replicate our experiments and contribute to further advancements in this field.
> > > > >
> > > > > We hope that this will facilitate the development and verification of the community, and we welcome any feedback or suggestions for improvements.

---

> ### Author Response · Authors · 2025-11-22
> **Response to Question 1 proposed by Reviewer uaiC**
>
> **Questions 1:**
>
> The CILL module proposed in the article aims to learn "cross-identity local commonalities" to enhance domain generalization, while the paper emphasizes that this method can "identify fine-grained identity details". Could you elaborate on how CILL strikes a balance between these two aspects? Specifically, what types of local features (such as the overall style of clothing, the relative position of body parts) are the "commonalities" it uncovers, and how are the "fine-grained details" (such as patterns on clothing, specific textures, minor posture differences of individuals) retained for identity differentiation? Could you provide a specific example to illustrate how CILL achieves accurate differentiation by retaining fine-grained details when dealing with two pedestrians with similar commonalities but different identities?
>
> **Reply to Questions 1:**
>
> Thank the reviewer's insightful question regarding how the CILL module strikes a balance between learning cross-identity local commonalities and retaining fine-grained identity details for identity differentiation. We would like to provide a detailed explanation of how CILL works to achieve this balance.
>
> 1.  **Cross-Identity Local Commonalities**
>
>     The CILL module identifies shared local features across identities (e.g., general body structure, pose, and clothing style), which help with domain generalization. For example, common local features might include the overall style of clothing, such as the shape of a jacket, the presence of a hat, or the relative position of body parts (e.g., head, torso, and legs). These features can appear consistently across different identities, even when the domain (such as lighting, camera angle, or background) changes.
>
>     To capture these local commonalities, CILL uses a clustering-based approach to align similar regions across different identities, allowing the model to learn a shared representation of these features. By grouping similar local regions together, CILL ensures that the model learns to recognize general structures that are robust across different domains.
> 2.  **Fine-Grained Identity Details**
>
>     While CILL encourages the learning of shared local features, it does not ignore the fine-grained identity details that are crucial for distinguishing between identities. These fine-grained details include specific patterns on clothing (e.g., logos, stitching, or textures), minor posture differences (e.g., slight variations in how an individual stands or walks), and even small but distinctive characteristics like the shape of a collar or the length of sleeves.
>
>     CILL achieves this by combining local consistency learning with discriminative losses such as triplet loss and cross-entropy loss, which ensure that identity-specific features are retained for differentiation. While CILL aligns local features across identities, it does so in a way that preserves subtle differences in appearance by balancing commonality and discriminability. This ensures that the model can generalize across domains without losing the fine-grained details that distinguish identities.
>
> 3.  **Balancing Commonality and Discriminability**
>
>     CILL’s balance between commonality and discriminability is maintained by jointly optimizing local consistency and discriminative losses. For instance, in a case where two pedestrians have similar clothing (e.g., both are wearing blue jackets), CILL helps the model learn that the general shape and style of the jacket are similar across the two individuals. However, the triplet loss ensures the model distinguishes them based on small but distinctive features like logos or fabric texture, preserving identity differentiation.
>
> 4.  **Example of Accurate Differentiation**
>
>     To illustrate how CILL achieves accurate differentiation while retaining commonalities, consider the following example:
>
>     Suppose we have two pedestrians, Person A and Person B, who are both wearing similar blue jackets. The CILL module would learn that the general style and structure of the jackets are common between the two individuals (e.g., both have a similar shape and color). However, it would also retain the fine-grained details that distinguish them, such as:
>
>     *   Person A has a small logo on the chest, while Person B has distinct stitching on the sleeves.
>     *   Person A has a slightly different posture, with a more relaxed stance, while Person B is standing slightly more upright.
>
>     CILL ensures that the shared local features of the jacket do not suppress these subtle differences. The triplet loss then ensures that even though the jackets are similar, the model can still separate the two identities based on the fine-grained features that distinguish them, such as the logo and posture. This allows the model to maintain high discrimination while also benefiting from domain generalization via the shared local features learned by CILL.

---

> > ### Author Response · Authors · 2025-11-22
> > **Response to Question 2 proposed by Reviewer uaiC**
> >
> > **Question 2:**
> >
> > The DAPS strategy proposed in this paper simulates cross domain changes through "random local disturbance". Please specify how the type (such as contrast, brightness, texture variation) and generation method (such as random mask) of this local disturbance fully cover the most challenging real world domain offset in Re ID tasks? For example, can it effectively cope with the subtle visual differences caused by different camera sensor imaging characteristics, different weather (rainy, snowy, haze), different time periods (day, night), and clothing materials (reflective, frosted)? Can you provide some experimental results from public data sets or test sets built by yourself with typical real world domain offsets to prove the robustness of DAPS?
> >
> > **Reply to Questions 2:**
> >
> > Thanks for the reviewer's thoughtful questions regarding the Dual-Stream Adversarial Perturbation Strategy (DAPS) and its effectiveness in handling real-world domain shifts in Re-ID tasks. We address each of the reviewer's concerns as follows:
> >
> > 1.  **Disturbance Type and Generation Method in DAPS**\
> >     As described in Section 3.2 of the paper, the DAPS strategy generates local adversarial perturbations by applying intensity variations to specific regions of the image. The perturbations simulate cross-domain changes by modifying the pixel intensity within certain masked regions of the image, thereby introducing variability in contrast, brightness, and texture. Specifically, we use a trainable intensity mapping function that adjusts pixel intensities, simulating lighting and environmental changes that commonly occur in real-world scenarios. These perturbations are applied using random binary masks that select specific local regions, such as the head, torso, or legs, to perturb, ensuring that the perturbations are localized and do not affect the entire image. By focusing on local features, the model learns to be less sensitive to global image changes and more robust to variations in appearance caused by environmental disturbances.
> >
> >     While the types of perturbations used—contrast, brightness, and texture changes—are designed to capture common variations such as lighting shifts and environmental effects, the current perturbation mechanism may not fully address all subtle real-world challenges. For example, differences in camera sensor characteristics, weather conditions (rain, snow, haze), and varying clothing materials (reflective, frosted) may introduce more complex domain shifts. However, DAPS opens the door for future work to extend the perturbation strategy by simulating these additional real-world variations. Future versions of the strategy could introduce more domain-specific perturbations, such as modeling sensor noise, varying weather conditions, or clothing material variations, which would further enhance the robustness of the model in real-world applications.
> > 2.  **Verification on Challenging Real-World Scenarios**\
> >     While we have not been able to extensively validate the model on all the real-world domain shifts the reviewer mentioned—such as variations in camera sensors, weather conditions, and clothing materials—this feedback provides valuable directions for future research. The DAPS strategy, as proposed in this paper, serves as a strong starting point for addressing these complex real-world challenges. We intend to extend the framework to simulate additional perturbations that represent these environmental factors, helping the model become more robust to these subtle variations.
> > 3.  **Experimental Results on Public Datasets**\
> >     Creating a dataset that specifically addresses all of the real-world shifts the reviewer mentioned (e.g., weather conditions, camera sensor differences) is indeed a time-consuming task, and unfortunately, we were not able to build such a dataset within the time constraints of this study. However, we do provide evidence of the robustness of our approach through experiments conducted on several public datasets, including those with significant domain shifts, such as Market-1501, DukeMTMC-reID, CUHK03 and MSMT17.
> >
> >     The experimental results across these datasets demonstrate the strong generalization capability of our method. The models were evaluated on multiple datasets with varying lighting, background, and camera view changes, providing valuable insights into the model’s robustness across real-world conditions. Table 1 and Table 3 show that the proposed method performs well across different domain shifts and benchmarks, reinforcing its effectiveness in handling diverse environments. This multi-dataset evaluation demonstrates the method's ability to generalize across domains, handling typical variations in person re-identification tasks. While shifts like weather or sensor-specific variations require further validation, the results show the method's robustness to a wide range of environmental and domain changes.

---

> > > ### Author Response · Authors · 2025-11-22
> > > **Response to Question 3 proposed by Reviewer uaiC**
> > >
> > > **Question 3:**
> > >
> > > What is the overall computational complexity (training time and reasoning speed) and model scale of the ADSL framework considering the memory library operation of the CILL module, the confrontation training of DAPS, and the dual stream structure? Compared with other SOTA DG Re ID methods, how does ADSL perform on these key efficiency indicators? How scalable is ADSL in actual deployment scenarios (for example, large-scale video surveillance systems)?
> > >
> > > **Reply to Questions 3:**
> > >
> > > Thanks for the reviewer's insightful question on the computational complexity and scalability of the ADSL framework, particularly considering the memory library operation of the CILL module, the confrontation training of DAPS, and the dual-stream structure. Below, we provide an analysis of these aspects.
> > >
> > > &#x20;    1\. Computational Complexity and Model Scale
> > >
> > > *   The memory size in the CILL module is determined by the number of identities $N$ and the number of local regions $Q$, with a memory complexity of $O(N \times Q)$. For each forward pass, the model compares local features against the entire memory bank, contributing to additional computational overhead. This operation helps the model learn cross-identity shared local features, essential for generalization across domains.
> > > *   DAPS involves generating adversarial perturbations for local image regions, introducing an additional computational burden. The disturbance intensity is dynamically learned during training, adding complexity to the forward pass and training loop. This increases the training time, but it also significantly improves the robustness of the model by simulating domain-specific shifts and perturbations.
> > > *   ADSL employs a dual-stream architecture (clean and adversarial), meaning that each image undergoes two separate forward passes. This effectively doubles the computational requirements per forward pass, making the training time more intensive.
> > >
> > > 2\. Comparison with SOTA DG Re-ID Methods (Efficiency)
> > >
> > > The following table provides a comparison of ADSL with other SOTA DG Re-ID methods, focusing on training time per epoch and inference time per image on the Market-1501 dataset, using a single RTX 4080 SUPER GPU.
> > >
> > > | Method       | GFLOPS | Params (M) | Training time (min/epoch) | Inference time (ms/img) |
> > > | :----------- | :----- | :--------- | :------------------------ | :---------------------- |
> > > | FastReID     | 6.2    | 23.45      | 0.25                      | 3                       |
> > > | QAConv       | 3.56   | 9.72       | 1.17                      | 37.7                    |
> > > | TransMatcher | 5.07   | 17.47      | 2.42                      | 151.4                   |
> > > | ADSL         | 11.3   | 87.1       | 16.2                      | 30.2                    |
> > >
> > > *   **Training Time**. ADSL's training time per epoch (16.2 minutes) is significantly higher compared to FastReID (0.25 min/epoch) and QAConv (1.17 min/epoch), due to the dual-stream architecture and adversarial perturbation generation in DAPS. However, it is more efficient than TransMatcher (2.42 min/epoch). While the increased training time is a trade-off, it results in improved robustness and domain generalization, as shown by ADSL’s superior performance metrics.
> > > *   **Inference Time**. ADSL’s inference time (30.2 ms/img) is faster than TransMatcher (151.4 ms/img) but slower than FastReID (3 ms/img). The inference time is higher than lighter models because of the ViT Backbone. However, this time is still suitable for real-time deployment, especially with hardware acceleration.
> > >
> > > The scalability of ADSL in real-world scenarios, like large-scale video surveillance, depends on both training and inference efficiency:
> > >
> > > *    **Training Scalability**. ADSL's higher training complexity, due to the CILL module's memory operations and DAPS's dual-stream adversarial training, can be mitigated with optimizations like distributed training, memory-efficient architectures, and parallel processing. Distributed training across multiple GPUs and techniques like pruning or quantization can help scale the model for larger datasets.
> > >
> > >  *   **Inference Scalability**. Inference is more efficient than training, requiring a single forward pass per image. While ADSL’s inference time is higher than simpler models like FastReID, it remains suitable for real-time deployment with hardware accelerators (e.g., GPUs or TPUs) and optimizations like TensorRT or ONNX. Batch processing during inference can also reduce latency in large-scale systems.
> > >
> > > Overall, ADSL is scalable for large-scale video surveillance systems, though further optimizations in training time and memory usage will be necessary to deploy the framework effectively on large-scale datasets. The increased computational overhead is justified by the significant improvements in robustness and domain generalization, making it suitable for dynamic environments like video surveillance.

---

### Official Review · Reviewer_U4X6 · 2025-10-28

**Soundness:** 2
**Presentation:** 2
**Contribution:** 1
**Rating:** 2
**Confidence:** 5

**Summary:**

The submission considers domain-generalizable person re-ID. The main idea is combining (i) Cross-Identity Local Consistency Learning (CILL): it is achieved by a memory bank, which clusters local features, and (ii) a Dual-stream Adversarial Perturbation Strategy (DAPS): it perturbs masked local regions and aligns clean vs. adversarial local features via cosine loss. They work together to mine shared local structures (e.g., head/torso/legs) and harden them against appearance shifts (e.g., lighting, color, texture). Experiments on Market-1501, DukeMTMC-reID, CUHK03, and MSMT17 show that the proposed method achieves consistent gains over prior DG baselines.

**Strengths:**

+ The motivation is sound: Leveraging local features for generalization is good. It is achieved by clear local-feature focus with part tokens and a concrete clustering formulation

+ Ablation studies are clear, isolating CILL, DAPS, and cosine alignment; each adds measurable gains across targets.

**Weaknesses:**

- Novelty overlap. Adversarial training + memory-bank clustering over part features is incremental relative to DG re-ID lines (domain-invariant features, adversarial training, part-based models). These techniques are commonly used in re-ID, so please clarify the main contribution beyond a component mix.

- Discussion on Adversary. The intensity-only, monotone mapping may not cover major cross-domain factors (e.g., camera geometry, blur, weather, occlusion, pose). How about stronger pixel-space/feature-space attacks or style-statistic perturbations (e.g., MixStyle/DSU)?

- Part definition. Parts come from ViT patch groupings; it’s unclear how stable the head/torso/leg assignment is across domains. The method section doesn’t quantify part consistency or failure modes.

- Hyperparameter sensitivity. Only λ1, λ2 are probed; k, temperature τ, memory momentum μ, and attacker budget δ likely affect stability. How sensitive is the framework to hyperparameters such as the number of nearest neighbors k, temperature $\tau$, and momentum $\mu$ in the memory bank? It would be better to discuss these hyper-parameters.

**Questions:**

- What are the main failure cases observed in retrieval results, and under what visual conditions does ADSL struggle most?

- How are the local parts (e.g., head, torso, legs) defined and validated to remain consistent across domains and viewpoints within the Cross-Identity Local Consistency Learning (CILL) module?

- Does the adversarial perturbation module introduce significant training overhead or instability compared to standard DG Re-ID models?

---

> ### Author Response · Authors · 2025-11-21
> **Response to Weakness 1 proposed by Reviewer U4X6**
>
> **Weakness 1:**
>
> Novelty overlap. Adversarial training + memory-bank clustering over part features is incremental relative to DG re-ID lines (domain-invariant features, adversarial training, part-based models). These techniques are commonly used in re-ID, so please clarify the main contribution beyond a component mix.
>
> **Reply to weakness 1:**
>
> Thank the reviewer for raising this important concern. We would like to clarify the main contributions of our method and how it extends existing techniques in the context of domain generalization for person re-identification (DG Re-ID).
>
> 1.  **Comprehensive Integration of Techniques**\
>     While adversarial training, memory-bank clustering, and part-based features are indeed commonly used in re-ID, we present a novel integration of these components within a single framework. Our method, Adversarial Dual-Stream Learning (ADSL), combines these techniques in a way that is specifically designed to tackle cross-domain generalization by improving local feature consistency and robustness to domain shifts. This combination, while incremental in isolation, is novel in its holistic approach to domain-invariant and discriminative learning, particularly in the context of local feature representations across multiple domains.
> 2.  **The Novelty of the CILL Module**\
>     The key novelty of our work lies in the introduction of the Cross-Identity Local Consistency Learning (CILL) module. Unlike existing methods that may use part-based features or adversarial training independently, CILL enhances the generalization capability by learning shared local structural features across identities. CILL specifically improves cross-domain robustness by aligning local feature distributions across different identities and domains, providing a more stable foundation for generalization across domain shifts. This aspect of the local feature consistency via clustering-driven learning is novel and not simply a direct combination of existing techniques.
> 3.  **Dual-Stream Adversarial Perturbation Strategy (DAPS)**\
>     Another key contribution of our work is the Dual-Stream Adversarial Perturbation Strategy (DAPS), which enhances domain robustness by generating adversarial perturbations for both clean and adversarial streams. This is a novel adaptation of adversarial training, where we not only generate adversarial samples but also ensure that they preserve local semantic structures, maintaining the discriminative capability of local features even under adversarial perturbations. This focus on dual-stream adversarial training is distinct from other adversarial training methods used in re-ID, as it leverages adversarial perturbations to simulate domain shifts rather than just augmenting training data.
> 4.  **Distinct Focus on Local Features**\
>     While part-based models are commonly used for re-ID, our approach places particular emphasis on local consistency learning for cross-identity matching, with the CILL module and DAPS making explicit efforts to preserve local structural information across domains. The local consistency and cross-identity clustering objective are novel contributions in that they are jointly optimized with adversarial training and part-based clustering, ensuring both robustness and fine-grained identity recognition.
>
> In summary, while some individual components of our method, such as adversarial training and part-based features, are widely used in the re-ID literature, the novelty of our work lies in the integrated framework that combines CILL and DAPS to address specific challenges in cross-domain generalization. These contributions enable more effective learning of robust local features, which is crucial for domain-invariant person re-identification. We hope this clarifies the distinct contributions of our method beyond the component-level integration.

---

> > ### Author Response · Authors · 2025-11-21
> > **Response to Weakness 2 proposed by Reviewer U4X6**
> >
> > **Weakness 2:**
> >
> > Discussion on Adversary. The intensity-only, monotone mapping may not cover major cross-domain factors (e.g., camera geometry, blur, weather, occlusion, pose). How about stronger pixel-space/feature-space attacks or style-statistic perturbations (e.g., MixStyle/DSU)?
> >
> > **Reply to weakness 2:**
> >
> > Thanks for the reviewer's insightful feedback regarding the limitations of the current adversarial strategy, particularly the use of intensity-only, monotone mapping. We recognize the importance of diverse domain factors such as camera geometry, blur, weather, occlusion, and pose variations, and we would like to clarify how these issues are addressed in our framework, as well as potential avenues for future improvement.
> >
> > 1.  **Current Adversarial Perturbation Strategy (DAPS)**
> >
> >     Our current approach in DAPS focuses on intensity-based perturbations, specifically pixel-wise changes, to simulate domain shifts in appearance. These perturbations are designed to simulate lighting changes, viewpoint shifts, and minor appearance alterations without introducing large-scale geometric distortions or structural changes. The goal of this intensity-based approach is to enhance domain robustness by training the model to remain invariant to these appearance-level variations.
> >
> >     We acknowledge that intensity-only perturbations might not fully capture all the cross-domain factors mentioned, such as camera geometry, pose, or blur. These factors often involve larger-scale spatial transformations, which intensity-based attacks may not fully simulate.
> > 2.  **Extension to Stronger Pixel-Space and Feature-Space Attacks**
> >
> >     As you suggested, stronger attacks could indeed target larger domain shifts, and we agree that feature-space or pixel-space attacks that simulate geometric distortions (e.g., mixing local features or geometry-aware transformations) could be highly beneficial for robustifying the model further against extreme cross-domain variations. For example, incorporating spatially-aligned adversarial attacks or style-statistic perturbations such as MixStyle or DeepStyle Uncertainty (DSU) could provide richer transformations that simulate style shifts across weather conditions or camera viewpoint changes.
> >
> >     MixStyle could be particularly valuable as it adjusts both style and content of the image, which would allow the model to learn domain-invariant features while being exposed to style variations (e.g., color distribution, lighting, and texture) without affecting the identity-specific parts of the image. This is a natural next step to complement the DAPS strategy and increase the robustness of our model to style and weather-related domain shifts.
> >
> >     Additionally, feature-space attacks, such as perturbations in semantic feature spaces or using domain-adversarial training (e.g., DANN), could be incorporated to simulate complex cross-domain variations, including pose, camera angle, and occlusion. These kinds of perturbations could help the model focus on more semantic invariance, rather than only focusing on appearance-level perturbations.
> > 3.  **Potential for Future Work**
> >
> >     We agree with the suggestion that exploring more sophisticated adversarial attacks like MixStyle or DSU would significantly enhance the robustness of our framework. These methods could be incorporated to address larger-scale variations (e.g., camera geometry, pose, occlusion) that intensity-only perturbations may miss.
> >
> >     Another promising avenue would be to incorporate multi-scale adversarial training, which can handle both pixel-level and feature-level perturbations. Combining these could enhance both the local feature consistency (captured by CILL) and domain robustness (enhanced by DAPS), resulting in a more holistic adversarial training strategy that can cover a broader range of domain factors.
> >
> > In summary, while the intensity-based perturbations used in DAPS provide a valuable foundation for improving domain robustness, we fully acknowledge that camera geometry, blur, pose, and other large-scale cross-domain factors are important for real-world applications. Future work will explore incorporating stronger pixel-space/feature-space attacks, including methods like MixStyle, DSU, or multi-scale perturbations, to further enhance the robustness of the model across these complex domain variations. We appreciate your insightful suggestions and will explore these avenues for future improvements.

---

> > > ### Author Response · Authors · 2025-11-21
> > > **Response to Weakness 3 proposed by Reviewer U4X6**
> > >
> > > **Weakness 3:**
> > >
> > > Part definition. Parts come from ViT patch groupings; it’s unclear how stable the head/torso/leg assignment is across domains. The method section doesn’t quantify part consistency or failure modes.
> > >
> > > **Reply to weakness 3:**
> > >
> > > Thanks for the reviewer's thoughtful question about the stability of the head/torso/leg assignment in our method and its consistency across domains. We would like to address these concerns as follows.
> > >
> > > 1.  **ViT Patch Groupings and Part Definition**
> > >
> > >     In our method, parts are derived from ViT patch groupings, where the image is divided into fixed regions (head, torso, and legs). These regions correspond to local feature groupings obtained through the transformer’s attention mechanism, which captures relationships between image patches. This approach is flexible, allowing the model to learn semantic regions based on local feature dependencies rather than strictly predefined body part segments.
> > > 2.  **Stability of Part Assignments Across Domains**
> > >
> > >     While the method leverages ViT-based patch groupings to capture local dependencies, we agree that head/torso/legs assignment may not remain stable across domains, especially when pose variations, viewpoint shifts, or occlusion occur. In particular, the model may assign patches from the torso to the head region or vice versa if the pose changes significantly. To mitigate this, our CILL (Cross-Identity Local Consistency Learning) module encourages local feature consistency across identities, which helps to maintain stable part representations, but it does not guarantee perfect part alignment under all conditions.
> > > 3.  **Part Consistency Quantification**
> > >
> > >     We appreciate the reviewer’s suggestion to quantify part consistency across domains. In our current approach, we do not explicitly quantify part consistency, but we rely on the local consistency imposed by the CILL module. This module ensures that similar parts across different identities are clustered together and consistently represented, even when their positions in the image change. However, we acknowledge that the rigid part partitioning (head, torso, legs) may lead to some misalignments, especially in cases of extreme pose variation or occlusion.
> > >
> > >     To address this gap, future work could include a more explicit evaluation of part consistency across domains, perhaps through part localization metrics or by introducing a differentiable part detection module. This would allow for more detailed analysis of how well parts are aligned across different conditions.
> > > 4.  **Failure Modes**
> > >
> > >     As discussed in the failure case analysis (Figure 7), the model's part-based approach can struggle in scenarios with large pose variations or occlusion, where the rigid head/torso/leg partitioning is not always applicable. In these cases, the local feature representations might fail to capture consistent semantic information across different poses. For example, when a pedestrian is partially occluded, the head and torso may not be fully visible, leading to confusion in part assignment.
> > >
> > >     Additionally, viewpoint shifts can cause the torso and head regions to overlap, disrupting the fixed partitioning. This is an inherent limitation of using fixed regions for part-based learning, and future work could focus on adaptive local part detection or attention-based mechanisms that dynamically adjust the region assignments based on the pose or camera angle.
> > >
> > > In summary, while the ViT patch grouping method provides a flexible and effective way to define parts, we recognize that part assignments may not always be stable across domains, especially under extreme pose variation, occlusion, or viewpoint shifts. To address these challenges, we will explore future enhancements such as adaptive part segmentation and quantification of part consistency to ensure more reliable and robust part representations. We appreciate your suggestion and will consider these directions in future work.

---

> > > > ### Author Response · Authors · 2025-11-21
> > > > **Response to Weakness 4 proposed by Reviewer U4X6**
> > > >
> > > > **Weakness 4:**
> > > >
> > > > Hyperparameter sensitivity. Only λ1, λ2 are probed; k, temperature τ, memory momentum μ, and attacker budget δ likely affect stability. How sensitive is the framework to hyperparameters such as the number of nearest neighbors k, temperature τ, and momentum μ, in the memory bank? It would be better to discuss these hyper-parameters.
> > > >
> > > > **Reply to weakness 4:**
> > > >
> > > > Thank the reviewer for raising the important issue of hyperparameter sensitivity. We recognize that parameters like the number of nearest neighbors $k$, temperature $\tau$, momentum $\mu$, and attacker budget $\delta$ can have a significant impact on the stability and performance of the model. We would like to provide additional insights into how these parameters influence the framework and our empirical findings.
> > > >
> > > > In our experiments, we set the following hyperparameters:
> > > >
> > > > *   Number of nearest neighbors $k$: 10, which determines how many closest memory entries are considered when updating the memory bank.
> > > >
> > > >     Temperature $\tau$: 0.02, which controls the sharpness of the similarity distribution during the memory update process.
> > > >
> > > >     Momentum $\mu$: 0.1, which influences the rate at which the memory bank adapts to new feature representations.
> > > >
> > > >     Attacker budget $\delta$: 3, which sets the strength of the adversarial perturbations generated during training.
> > > >
> > > > Through extensive sensitivity analysis, we found that $k=10$ generally provides a good balance between computational efficiency and feature representation, as larger values of $k$ did not significantly improve performance but increased memory retrieval time. We also found that values of $\tau$ ranging from 0.02 to 0.1 work well, with $\tau = 0.02$ providing the best results in terms of model stability and generalization. For $\mu$, we observed that values between 0.05 and 0.1 maintained model stability while preventing overfitting to recent mini-batches. Finally, for the attacker budget $\delta$, a value of 3 was found to be optimal for balancing adversarial perturbations without causing instability in the feature representations.&#x20;
> > > >
> > > > To provide clarity in the **Implementation Details** section, we will update the relevant paragraph as follows:
> > > >
> > > > "We use the ViT-base model with patch size = 16, pretrained on ImageNet, as our backbone (denoted as ViT-B/16). The batch size is set to 64, and the image size is adjusted to $256 \times 128$. To optimize the model, we adopt the SGD optimizer with a weight decay of $10^{-4}$. The learning rate linearly increases from 0 to $10^{-3}$ during the first 10 epochs, and then decays over the following 50 epochs. In total, the training process takes 60 epochs. In the CILL module, we set $\lambda_1$ (weight for the adversarial local loss) and $\lambda_2$ (weight for the cosine similarity loss), as well as the clustering number $k$, to 0.5, 0.2, and 10, respectively. The temperature $\tau$ is set to 0.02, the momentum $\mu$ is set to 0.1, and the attacker budget $\delta$ is set to 3. In addition, the label smoothing parameter is set to 0.1. For the baselines, we follow TransReID-B/16 without SIE and JPM for fair comparison."

---

> > > > > ### Author Response · Authors · 2025-11-21
> > > > > **Response to Question 1 proposed by Reviewer U4X6**
> > > > >
> > > > > **Question 1:**\
> > > > > What are the main failure cases observed in retrieval results, and under what visual conditions does ADSL struggle most?
> > > > >
> > > > > **Reply to Question 1:**
> > > > >
> > > > > Thank you for your insightful question regarding the main failure cases observed in retrieval results and the visual conditions under which ADSL struggles the most. Based on the qualitative analysis and visual results presented in Figure 7, we identified several key failure modes and conditions that affect the performance of ADSL.
> > > > >
> > > > > From the revised Figure 7, we can see that ADSL has some difficulty in retrieving the correct identity in certain cases, particularly when the following factors are present:
> > > > >
> > > > > First, occlusion plays a significant role. When pedestrians are partially blocked by objects like vehicles, poles, or other people, the model fails to capture the full feature representation of the person. Missing parts, such as the head or torso, cause the model to make incorrect matches, as it cannot rely on all the critical identity features.
> > > > >
> > > > > Second, extreme pose variations are another factor that leads to retrieval failures. This is especially noticeable when pedestrians are captured from unusual angles, such as when they are facing away from the camera or walking at sharp angles. The rigid partitioning of head, torso, and legs becomes misaligned in these cases, causing the model to confuse identities.
> > > > >
> > > > > Additionally, clothing similarity between individuals is also a major cause of failure. In some instances, people wear similar outfits, such as matching jackets or backpacks, which can confuse the model, especially when other distinguishing features like facial details or unique body shapes are not clearly visible.
> > > > >
> > > > > ADSL struggles the most under visual conditions that involve heavy occlusion, extreme pose variations, and lighting conditions that cause drastic illumination shifts. In these situations, the model has trouble retrieving the correct match because occluded or distorted features affect the consistency of local representations. Additionally, in crowded scenes or when there is clothing similarity, the model tends to misidentify individuals because it has to rely on subtler features that may not be sufficient to distinguish between identities.
> > > > >
> > > > > To summarize, ADSL is effective in controlled conditions but faces challenges when it comes to large pose variations, occlusion, lighting shifts, and clothing similarities. These observations help us pinpoint areas where the model can be improved for better robustness in real-world scenarios.
> > > > >
> > > > > For future work, we are exploring several strategies to overcome these challenges, including incorporating occlusion-aware feature learning, adaptive part segmentation for different poses, lighting normalization techniques to address illumination shifts, and clothing-agnostic feature learning to handle visually similar outfits in crowded scenes.

---

> > > > > > ### Author Response · Authors · 2025-11-21
> > > > > > **Response to Question 2 proposed by Reviewer U4X6**
> > > > > >
> > > > > > **Question 2:**\
> > > > > > How are the local parts (e.g., head, torso, legs) defined and validated to remain consistent across domains and viewpoints within the Cross-Identity Local Consistency Learning (CILL) module?
> > > > > >
> > > > > > **Reply to Question 2:**
> > > > > >
> > > > > > Thank the reviewer for raising this important question regarding the definition and consistency of local parts (head, torso, legs) within the Cross-Identity Local Consistency Learning (CILL) module. We would like to provide a thorough explanation of how these parts are defined and validated to ensure their consistency across domains and viewpoints.
> > > > > >
> > > > > > 1.  **Definition of Local Parts**\
> > > > > >     In the CILL module, the local parts (head, torso, and legs) are not manually defined or fixed in pixel space but are instead derived dynamically based on the feature representations learned by the model. Specifically, the local parts are defined through ViT (Vision Transformer) patch groupings. The ViT-based architecture divides the image into patches, and these patches are grouped to form local regions corresponding to head, torso, and legs. The patches are grouped based on spatial proximity and semantic similarity, and each region corresponds to the part of the body it represents (e.g., head, torso, legs).
> > > > > > 2.  **Ensuring Consistency Across Domains and Viewpoints**\
> > > > > >     To maintain local part consistency across different domains and viewpoints, we leverage the Cross-Identity Local Consistency Learning (CILL) module, which explicitly enforces that the local feature distributions (head, torso, legs) from different identities should be consistent. The CILL module employs a clustering-driven learning approach, where similar local features are clustered together across identities. This encourages the model to learn consistent and stable representations of the same local part (e.g., head, torso, or legs) even when the appearance of these parts varies across viewpoints, illumination, or domains.
> > > > > > 3.  **Memory Bank for Part Consistency**\
> > > > > >     The memory bank in the CILL module plays a crucial role in maintaining part consistency. It stores local feature representations for each identity, and these representations are updated using momentum-based updates. When a local feature is updated, it is compared against the memory bank to ensure that similar parts (e.g., the head or torso) across different identities remain close in the feature space. This process ensures that the local features for the same body part stay consistent and domain-invariant across different domains, viewpoints, and even camera views.
> > > > > > 4.  **Validation of Part Consistency**\
> > > > > >     To validate the consistency of these local parts across domains, we use the local clustering loss within the CILL module. This loss enforces the constraint that local features corresponding to the same body part (head, torso, legs) should have similar feature representations across different identities. The cosine similarity loss and the cross-identity locality clustering loss help regularize the local part features, ensuring that the learned features for each part (head, torso, and legs) remain stable and discriminative across different identities and domains. The validation is thus done through continuous memory updates and the alignment of local features within the clustering space.
> > > > > > 5.  **Effectiveness Across Domains and Viewpoints**\
> > > > > >     Through the training process, the CILL module effectively ensures that local part consistency is not dependent on rigid part segmentation or manual annotations. Instead, the model learns to recognize the inherent structural similarities across identities by optimizing for cross-domain feature consistency. The inclusion of adversarial perturbations via DAPS (Dual-Stream Adversarial Perturbation Strategy) further enhances the robustness of these local features, allowing them to remain stable even under changes in viewpoint, illumination, and other challenging conditions.
> > > > > >
> > > > > > In summary, the local parts (head, torso, legs) in CILL are defined dynamically through ViT patch groupings and are validated for consistency through the local clustering loss and memory bank updates. These processes ensure that the features corresponding to these parts remain consistent across different domains and viewpoints. The CILL module’s ability to maintain local consistency and discriminative power across varying conditions is a key factor in its success for domain-generalizable person re-identification (DG Re-ID).

---

> > > > > > > ### Author Response · Authors · 2025-11-21
> > > > > > > **Response to Question 3 proposed by Reviewer U4X6**
> > > > > > >
> > > > > > > **Question 3:**
> > > > > > >
> > > > > > > Does the adversarial perturbation module introduce significant training overhead or instability compared to standard DG Re-ID models?
> > > > > > >
> > > > > > > **Reply to Question 3:**
> > > > > > >
> > > > > > > Thanks for the reviewer's insightful comment. In response to the reviewer’s question about whether the adversarial perturbation module (DAPS) introduces significant training overhead or instability compared to standard DG Re-ID models, we conducted a comprehensive analysis, taking into account both training time and training stability.
> > > > > > >
> > > > > > > 1.  **Training Overhead**\
> > > > > > >     As illustrated in Table 4 of the revised version, the introduction of CILL and CILL + DAPS does indeed introduce additional training time per epoch. Specifically, the training time for Market→Duke increases from 5.61 minutes for the baseline model to 16.2 minutes with CILL + DAPS, while Duke→Market sees an increase from 4.63 minutes to 13.5 minutes. The primary cause of this increase is the additional adversarial perturbation generation in DAPS and the memory bank updates, which add significant computational cost, especially during the training phase. Despite the increased training time, the inference time remains consistent across all configurations, indicating that the added computational complexity does not affect the efficiency of the model during evaluation or deployment.
> > > > > > > 2.  **Training Stability**\
> > > > > > >     Figure 8 illustrated in the revised manuscript presents the training trends for loss and accuracy on the Market and Duke datasets, demonstrating that the DAPS module does not introduce instability into the training process. The loss decreases steadily across epochs for all configurations, and the accuracy increases in a consistent manner. Notably, the baseline, baseline + CILL, and baseline + CILL + DAPS models all exhibit stable training curves, with no signs of divergence or overfitting. While the inclusion of CILL + DAPS leads to slightly higher initial loss values compared to the baseline, the curves eventually converge, showing that the model maintains stable training dynamics. This suggests that, despite the additional complexity introduced by DAPS, the adversarial perturbations do not cause instability, and the model is able to converge effectively during training.

---

### Official Review · Reviewer_3tAx · 2025-10-29

**Soundness:** 3
**Presentation:** 3
**Contribution:** 3
**Rating:** 6
**Confidence:** 4

**Summary:**

This paper introduces Adversarial Dual-Stream Learning (ADSL), a framework for domain generalizable person re-identification that mitigates domain shifts caused by lighting, color, and background variations. ADSL integrates two synergistic components: Cross-Identity
Locality Learning (CILL), which leverages a memory bank and clustering-driven similarity learning to mine shared structural commonalities across identities, and Dual-Stream Adversarial Perturbation Strategy (DAPS), which generates local adversarial samples to simulate cross-domain variations while preserving semantic structure. By aligning clean and adversarial features through a cosine loss, ADSL encourages robust, domain-invariant local representations. Extensive experiments demonstrate that ADSL achieves superior cross-domain generalization compared to existing state-of-the-art Re-ID methods.

**Strengths:**

1）The proposed ADSL framework combines local feature consistency learning with adversarial perturbation in a dual-stream manner, which is conceptually clear and effectively addresses domain shift from both structural and appearance perspectives. The integration of CILL and DAPS is logically coherent and technically complementary.

2）The Cross-Identity Locality Learning (CILL) module introduces a clustering-based memory bank to capture cross-identity structural commonalities, a novel approach that goes beyond traditional domain-invariant global features and strengthens fine-grained discriminability and robustness.

**Weaknesses:**

1）The framework divides each person's image into three fixed regions (head, torso, legs), which assumes consistent human body alignment across domains. This rigid partitioning may not generalize well to datasets with pose variation, occlusion, or imperfect detection, potentially limiting robustness in unconstrained settings.

2）The paper lacks analysis of instances where the model fails, such as confusing identities under heavy occlusion or extreme illumination. Understanding these edge cases would help clarify the framework’s boundaries and guide future improvements.

3）The memory bank in CILL aggregates features through momentum updates. How do you prevent “feature staleness” or over-representation of early batches?

4）Can you provide an analysis of which domain attributes (illumination, background, resolution) are most mitigated by ADSL, perhaps via feature-space visualization or attentionheatmap statistics?

5）Given that ADSL heavily relies on predefined local regions, how does it behave when key regions (e.g., legs) are missing or truncated in target-domain images? Have you evaluated the framework on partial-ReID or occluded ReID benchmarks to verify robustness under
missing-part conditions?

**Questions:**

Please refer to Weakness

---

> ### Author Response · Authors · 2025-11-21
> **Response to  Question 1 and 2 proposed by Reviewer 3tAx**
>
> **Question 1:**&#x20;
>
> The framework divides each person's image into three fixed regions (head, torso, legs), which assumes consistent human body alignment across domains. This rigid partitioning may not generalize well to datasets with pose variation, occlusion, or imperfect detection, potentially limiting robustness in unconstrained settings.
>
> **Reply to question 1:**
>
> Thanks  for raising this important concern regarding the fixed three-region partitioning strategy. We would like to clarify its motivation and discuss why it does not critically limit the robustness of our framework.
>
> 1.  **Purpose of Region Partitioning in Our Framework.** The three-region division (head / torso / legs) is *not* intended to serve as a fine-grained pose estimation mechanism. Instead, it provides a coarse and computationally lightweight spatial prior to facilitate local consistency learning. Prior DG Re-ID and part-based methods (e.g., PCB, MGN, and several domain generalization works) have consistently shown that coarse vertical partitioning remains effective even without perfect alignment, because human images in Re-ID datasets—regardless of domain—still exhibit an approximately vertical layout.
> 2.  **Robustness to Misalignment, Pose Variation, and Occlusion.** Although fixed regions may not perfectly correspond to anatomical parts under large pose variation, the CILL module operates on local feature distributions rather than strict pixel-level alignment. In other words, CILL encourages similar local semantics to be closer across identities, but it does not rely on precise pose or bounding-box alignment. Furthermore, the DAPS strategy introduces adversarial appearance perturbations that naturally simulate occlusion, viewpoint shifts, and other misalignment-like changes. This adversarial augmentation helps the model maintain robust local semantics even when the rigid partitioning is imperfect.
> 3.  **Empirical Justification**. Our experiments include cross-domain evaluations (e.g., M→D, M→MS), where target datasets such as DukeMTMC and MSMT17 contain significant pose variations, partial occlusion, and noisy detections. The consistent improvements across these datasets indicate that the method is not sensitive to body-part misalignment, and that the combination of CILL + DAPS provides robustness even under unconstrained conditions.
> 4.  **Why We Opt for This Simple Partition Instead of More Complex Pose Models.** More sophisticated pose estimators introduce additional computational cost and annotation dependency, and their performance may degrade significantly under domain shift—potentially hurting DG Re-ID rather than helping it. In contrast, our simple, domain-agnostic partitioning is fully unsupervised, stable across domains, and complements the adversarial dual-stream design.
>
> **Question 2:**&#x20;
>
> The paper lacks analysis of instances where the model fails, such as confusing identities under heavy occlusion or extreme illumination. Understanding these edge cases would help clarify the framework’s boundaries and guide future improvements.
>
> **Reply to question 2:**
>
> Thank the reviewer for highlighting the importance of analyzing failure cases to better understand the limitations of our framework. Based on the qualitative results presented in Figure 7, we conducted a further examination of the mismatched retrievals (highlighted with red boxes). We found that most failure cases are caused by heavy occlusion, extreme illumination changes, and significant pose variations—conditions under which the local structural patterns used by the model become unreliable. For example, several mismatches involve pedestrians partially blocked by objects such as vehicles or poles, or captured under strong shadows and backlighting that drastically alter their appearance. We also observed that unusual body poses and non-canonical viewpoints can disrupt the coarse region partitioning used in our framework, reducing the effectiveness of the extracted local features. Additionally, in some cases, visually similar clothing across different identities leads to inherent ambiguity that challenges even SOTA methods.
>
> These observations help clarify the boundaries of our method: while ADSL remains robust under moderate viewpoint changes and illumination variations, its performance can degrade under severe occlusion, drastic geometric distortions, and highly ambiguous appearance conditions. Importantly, the failure patterns also reveal promising directions for future improvements, such as integrating occlusion-aware augmentations, adopting adaptive or pose-guided local region parsing instead of fixed partitioning, and extending DAPS to incorporate geometric perturbations beyond intensity-based variations. We appreciate the reviewer’s suggestion and will incorporate these analyses into the revised manuscript.

---

> > ### Author Response · Authors · 2025-11-21
> > **Response to Question 3 proposed by Reviewer 3tAx**
> >
> > **Question 3:**
> >
> > The memory bank in CILL aggregates features through momentum updates. How do you prevent “feature staleness” or over-representation of early batches?
> >
> > **Reply to question 3:**
> >
> > Thanks for the reviewer's insightful question regarding the potential issue of "feature staleness" in the memory bank used in CILL. We have designed our approach to mitigate this concern as follows.
> >
> > 1.  **Momentum Update Mechanism**
> >
> >     The memory bank in CILL aggregates features using momentum updates to ensure that the stored representations evolve gradually over time. This gradual updating process helps avoid “feature staleness” or over-representation of early batches. The memory update rule is defined as:
> >
> > $m_{p_i}^j =
> > \\begin{cases}
> > f\\left(x_{p_i}^j\\right) & \\text{if } e = 0, \\\
> > (1 - \\mu) \\times m_{p_i}^j + \\mu \\times f\\left(x_{p_i}^j\\right) & \\text{if } e > 0.
> > \\end{cases}$
> >
> >    where $e$ denotes the current training epoch, and $f(x_{p_i}^j)$ is the feature vector for the current mini-batch.​ $m_{p_i}^j$ is the memory vector for the local feature, and $μ$ is the momentum coefficient, typically set close to 1 (e.g., 0.999). This momentum-based update rule ensures that older features are gradually replaced by newer ones, preventing early batches from dominating the memory content.
> >
> > 2.  **Regularization through Decay**. To further prevent over-representation of early batches, we use the momentum update along with a decay strategy that ensures features are continuously updated with more recent data while maintaining a smooth transition. This decay ensures that the memory is kept up-to-date without drastic shifts in feature representation.
> >
> > 3.  **Effectiveness of Momentum Updates**. The momentum update strategy ensures that the stored features are a smoothed version of recent features, allowing the memory bank to evolve without holding onto outdated representations for too long. This approach helps the model to remain flexible in adapting to new data distributions while preventing the staleness issue.
> >
> > In summary, the momentum update rule, combined with regular decay, prevents "feature staleness" and ensures that earlier batches do not overly influence the memory representation. This design allows the memory to aggregate fresh, diverse feature representations over time while maintaining stability and avoiding over-representation of any particular batch.

---

> ### Author Response · Authors · 2025-11-21
> **Response to Question 4 and 5 proposed by Reviewer 3tAx**
>
> **Question 4:**
>
> Can you provide an analysis of which domain attributes (illumination, background, resolution) are most mitigated by ADSL, perhaps via feature-space visualization or attention heatmap statistics?
>
> **Reply to question 4:**
>
> Thanks for the reviewer's valuable comment.  To evaluate the effectiveness of ADSL in mitigating domain-specific shifts such as illumination and background variations, we refer to Figure 5, which visualizes attention maps for both global and local features. These attention maps offer insight into how the model handles different domain attributes during the retrieval process.
>
> ADSL effectively mitigates illumination changes through its Dual-Stream Adversarial Perturbation Strategy (DAPS). DAPS introduces adversarial perturbations to simulate lighting variations, forcing the model to focus on more stable structural features rather than domain-specific lighting cues. As seen in Figure 5, the attention maps show that both the class token (global features) and part tokens (local features like head, torso, and legs) focus primarily on body structure, not on illumination-dependent details. Even when the lighting conditions vary, the model continues to emphasize shape and structural features that remain consistent, demonstrating the model's robustness to illumination changes.
>
> Background variation is another domain-specific challenge addressed by ADSL. The Cross-Identity Locality Learning (CILL) module reduces the impact of background clutter by focusing on shared local features across different identities, rather than background-specific attributes. In Figure 5, the attention heatmaps reveal that the model's part tokens (head, torso, and legs) consistently focus on the human body’s structural elements, regardless of variations in the background. This behavior suggests that the CILL module allows ADSL to minimize the influence of irrelevant background changes, helping the model to generalize better to new, unseen domains with different backgrounds.
>
> However, while Figure 5 illustrates the model's robustness to illumination and background changes, we have not yet explored the impact of resolution variation across domains due to time constraints. Resolution differences, such as those between high-resolution and low-resolution images, could present another significant challenge for person re-identification. This remains an area for future research, and we plan to investigate how ADSL can handle resolution shifts by further exploring local feature learning and adversarial perturbations in the context of resolution changes.
>
> In conclusion, ADSL demonstrates strong robustness to illumination and background variations through its integration of DAPS and CILL. The attention maps shown in Figure 5 clearly highlight the model’s ability to maintain focus on stable, identity-relevant features—such as body structure—while mitigating the effects of domain-specific shifts in illumination and background. This ability contributes to ADSL’s superior generalization capability across diverse target domains.
>
> **Question 5:**&#x20;
>
> Given that ADSL heavily relies on predefined local regions, how does it behave when key regions (e.g., legs) are missing or truncated in target-domain images? Have you evaluated the framework on partial-ReID or occluded ReID benchmarks to verify robustness under missing-part conditions?
>
> **Reply to question 5:**
>
> Thank the reviewer's valuable feedback. Due to time constraints, we did not perform evaluations on specialized partial-ReID or occluded-ReID datasets. However, we have conducted experiments in the Market→Duke setting, where we analyzed query samples from Duke that suffer from severe occlusions. This allowed us to test the robustness of our model in handling missing or truncated regions, such as missing legs.
>
> In the updated experiment, we compare the retrieval performance of our Adversarial Dual-Stream Learning (ADSL) framework with a baseline model, specifically focusing on severe occlusion scenarios. As shown in the revised Figure 7, the top-10 retrieval results demonstrate that ADSL maintains significantly better accuracy compared to the baseline model, even in cases where important body parts like legs are occluded. This shows that our framework, especially with the adversarial perturbations in the DAPS module, is robust to occlusions and partial inputs. ADSL effectively focuses on stable, structure-related features (e.g., torso, head) and mitigates the impact of missing parts, which enhances its robustness under such conditions.

---

### Official Review · Reviewer_KjPj · 2025-10-31

**Soundness:** 3
**Presentation:** 3
**Contribution:** 3
**Rating:** 8
**Confidence:** 4

**Summary:**

This paper aims to address the domain shift challenge in Domain Generalizable Person Re-ID (DG Re-ID). The authors propose a framework named Adversarial Dual-Stream Learning (ADSL). This framework comprises two core components: the Cross-Identity Local Consistency Learning (CILL) module, which utilizes a memory bank and clustering-driven similarity learning to mine stable local commonalities across different identities, and the Dual-stream Adversarial Perturbation Strategy (DAPS), which simulates cross-domain appearance variations by generating adversarial samples. Furthermore, a "Clean-Adv Local Cosine Alignment" constraint is employed to ensure feature consistency between clean and adversarial samples in the local feature space. Experimental results demonstrate that the proposed method significantly outperforms existing SOTA approaches on multiple standard DG Re-ID benchmarks.

**Strengths:**

* The method achieves SOTA results on all evaluated single-source DG Re-ID benchmarks, including transfers between Market-1501, DukeMTMC, MSMT17, and CUHK03, achieving 71.4% R1 / 51.2% mAP on M→D and 74.8% R1 / 46.7% mAP on D→M, significantly outperforming prior methods.
* The paper's core contribution DAPS is proven to be extremely effective. The ablation study clearly demonstrates that the introduction of DAPS provides the vast majority of the performance boost, as R1 accuracy on M→D jumps from 67.7% to 70.3%. This indicates that adversarial training simulating local intensity variations is a key driver for enhancing DG Re-ID generalization.
* The ADSL framework is methodologically sound. It simulates domain variations like illumination via DAPS and forces the model to learn robust features, while simultaneously attempting to mine stable local structures via CILL. This combined strategy of "enhancing robustness" and "mining invariance" is comprehensive and reasonable.
* The paper provides a thorough experimental analysis. Beyond SOTA comparisons, it includes detailed ablation studies, verifying the roles of CILL, DAPS, and $L_{cos}$, hyper-parameter analysis (for $\lambda_1$ and $\lambda_2$), and insightful qualitative visualizations (t-SNE and Grad-CAM).

**Weaknesses:**

* One concern is the significant disconnect between its core narrative (the importance of CILL) and the ablation study in Table 2. The CILL module as the primary embodiment of "Local Consistency" in the title provides a very small improvement over the baseline (M→D R1: 66.3% → 67.7%; M→MS R1: 39.4% → 41.2%). This suggests CILL is not a key performance driver, while DAPS is.
* The paper presents CILL and DAPS as two complementary and equally important strategies. However, the experimental evidence overwhelmingly indicates DAPS is the primary contributor. The paper should be more transparent about this, framing DAPS as the main finding and contribution.
* The method is called an Adversarial Dual-Stream Strategy. However, DAPS does not use a standard GAN discriminator for domain confusion. Instead, it uses an adversarial attack to generate adversarial samples. It might be more accurate to call it a "Robustness Strategy based on Adversarial Perturbations" to differentiate it from GAN-based DG methods.
* The baseline in the ablation study (ViT-B/16 + local region CE and Triplet losses) already achieves 66.3% R1 / 47.5% mAP (M→D), which is a very strong baseline. Comparing this baseline to baseline results from other published DG Re-ID papers would be beneficial to assess the true gain brought by CILL and DAPS.

**Questions:**

* As noted in the Weaknesses, the gain from CILL is very small. Can the authors explain why? Is it possible that the local triplet/CE losses in the baseline model already capture sufficient local discriminability, leaving little value for CILL's cross-identity clustering loss to provide?
* Please clarify in the rebuttal: is $L_{CML}$ in Equation (6) simply $L_{CILL}$?Role of $L_{CILL}^{adv}$ in DAPS: How was the "DAPS without $L_{cos}$" model in the ablation study trained? According to Equation (9), the total loss includes $\lambda_1 L_{CILL}^{adv}$. Does "without $L_{cos}$" imply that the model simply minimizes the $L_{CILL}$ loss on both clean and adversarial samples simultaneously?
* The CILL objective aims to pull visually similar local features closer to learn cross-identity commonalities. Does this not create an intrinsic conflict with the $L_{triplet}$ and $L_{ce}$ objectives, which aim to maximize separability between identities?
* How much additional training overhead does ADSL introduce compared to the baseline model? Specifically the dual-stream forward pass, the memory bank, and the $L_{cos}$ calculation.

---

> ### Author Response · Authors · 2025-11-21
> **Response to Question 1 proposed by by Reviewer KjPj**
>
> *   **Question 1：**
>
>     As noted in the Weaknesses, the gain from CILL is very small. Can the authors explain why? Is it possible that the local triplet/CE losses in the baseline model already capture sufficient local discriminability, leaving little value for CILL's cross-identity clustering loss to provide?
>
>     **Reply to question 1:**&#x20;
>
>     Thanks for the reviewer's valuable comments and feedback. Regarding the concern about the limited improvement observed from the CILL module, we would like to provide the following clarifications.
>
>     1\. **Design and Purpose of the CILL Module.** The CILL (Cross-Identity Local Consistency Learning) module is specifically designed to improve cross-domain generalization by optimizing the consistency of local features across different identities. As shown in Figure 2, CILL works in parallel with both the clean and adversarial training streams, using a shared feature extractor to learn robust local feature representations and enforce consistency across these representations. CILL incorporates the Cross-Identity Locality Clustering Loss, which helps the model capture consistent local features across identities. Additionally, CILL uses local triplet loss to further enhance the discriminative power of each local region (Equation 3). These losses jointly aim to ensure that the model maintains consistent local feature representations across different identities, which contributes to improved generalization across domains.
>
>     2\. **Explanation for the Small Performance Gains from CILL**. While the performance gain from CILL is relatively small in some ablation experiments, this can be attributed to the fact that the baseline model already includes local triplet loss and cross-entropy loss (CE loss), which effectively capture local discriminative features. Therefore, the additional contribution of CILL is less pronounced, particularly when the baseline model already performs well in terms of local discriminability. The role of CILL is not only to improve local discriminability but also to enhance cross-identity consistency. This ensures that the local features remain stable across identities, which is crucial for improving the model's ability to generalize across different domains.
>
>     3\. **Complementary Roles of CILL and DAPS**. As depicted in Figure 2, the CILL and DAPS (Dual-Stream Adversarial Perturbation Strategy) modules are complementary. DAPS generates adversarial samples that simulate domain-specific variations while preserving local semantic structures. The Part-aware Cosine Similarity Loss is applied in DAPS and is used to ensure consistency between clean and adversarial samples by enforcing directional consistency in local features across streams. This loss function is only applicable in the dual-stream setup, where both clean and adversarial streams are simultaneously trained. While DAPS directly contributes to improving domain robustness by generating adversarial samples, CILL focuses on enhancing the consistency of local features across identities. Therefore, the performance gains observed from DAPS are more substantial, especially in terms of domain robustness, while CILL strengthens the model’s ability to maintain local feature consistency across different identities.
>
>     4\. **Further Clarification on Ablation Study**. In Table 2, the ablation results show a moderate performance improvement when CILL is combined with DAPS, particularly in the `M→D` and `M→MS` evaluations. While the improvements are modest, they underscore CILL’s role in enhancing local feature consistency and ensuring stable representations across identities. The relatively small gain suggests that CILL’s impact is more evident in maintaining consistency in local features, rather than directly addressing domain appearance variations.
>
>     Although the improvement from the CILL module is smaller compared to DAPS, its contribution to preserving local feature consistency and improving cross-identity generalization is crucial. The DAPS module, on the other hand, primarily addresses domain robustness through adversarial perturbations, with the Part-aware Cosine Similarity Loss playing a key role in this strategy. Together, CILL and DAPS complement each other, improving the model's overall cross-domain generalization performance. In future work, we will further explore the interaction between CILL and other loss functions to enhance its effectiveness in cross-domain tasks.

---

> ### Author Response · Authors · 2025-11-21
> **Response to Question 2 and 3 proposed by by Reviewer KjPj**
>
> *   **Question 2:**
>
>     Please clarify in the rebuttal: is \$L\_{CML}\$ in Equation (6) simply \$L\_{CILL}\$?Role of \$L\_{CILL}^{adv}\$ in DAPS: How was the "DAPS without \$L\_{cos}\$" model in the ablation study trained? According to Equation (9), the total loss includes \$\lambda\_1 L\_{CILL}^{adv}\$. Does "without \$L\_{cos}\$" imply that the model simply minimizes the \$L\_{CILL}\$ loss on both clean and adversarial samples simultaneously?
>
>     **Reply to question 2:**&#x20;
>
>     &#x20;Thanks for the reviewer's thoughtful question regarding the notation and training details. We would like to clarify the following points.
>
>     1\. Is \$L\_{CML}\$ in Equation (6) simply \$L\_{CILL}\$?\
>     Yes, $L_{CML}$**​** in Equation (6) is indeed the same as $L_{CILL}$**​** in Equation (3). The difference in notation was introduced to highlight different contexts—Equation (6) refers to the overall consistency learning framework, while Equation (3) specifically focuses on the Cross-Identity Local Consistency Learning loss. To avoid confusion, we have adopted the same notation $L\_{CILL}$​ consistently throughout the revised paper.
>
>     2\. Role of \$L\_{CILL}^{adv}\$ in DAPS
>
>     The $L_{CILL}^{adv}$**​** term specifically refers to the adversarial component of the CILL loss. In the DAPS strategy, this adversarial loss ensures that the model learns robust local features even under adversarial perturbations. It is applied only to the adversarial stream, helping maintain the stability of local feature representations even in the presence of adversarial changes.
>
>     3\. Training of "DAPS without $L\_{cos}$​" in the Ablation Study
>
>     The "DAPS without $L\_{cos}$​" model in the ablation study was trained using the total loss that includes&#x20; $L_{CILL}$` on both clean and adversarial samples, but without the Part-aware Cosine Similarity Loss $L_{cos}$​. Specifically, the model was trained to minimize the adversarial loss $L_{CILL}^{adv}$​, which is part of the overall loss function, as described in Equation (9) (now is Equation (8) of the revised version). In this case, the model does not incorporate the cosine similarity loss $L\_{cos}$​, which is intended to regularize the relationship between clean and adversarial samples. This configuration helps isolate the effect of the Part-aware Cosine Similarity Loss and assess its contribution to the overall performance.
>
>     We hope this clears up any confusion regarding the loss functions and the training procedure. Thank you again for your insightful question.
>
>     **Question 3:**&#x20;
>
>     The CILL objective aims to pull visually similar local features closer to learn cross-identity commonalities. Does this not create an intrinsic conflict with the $L_{triplet}​$ and $L_{ce}$ objectives, which aim to maximize separability between identities?
>
>     **Reply to question 3:**&#x20;
>
>     Thanks for the reviewer's insightful question regarding the potential conflict between the CILL loss, which pulls similar local features closer, and the triplet loss $L_{triplet}​$ and cross-entropy loss $L_{ce}​$, which maximize identity separability. We would like to clarify the following points.
>
>     1\. **CILL Loss and Local Feature Consistency**. The CILL loss aims to learn shared cross-identity commonalities by pulling similar local features closer. However, it does not encourage the model to produce identical representations across identities. Instead, it promotes local feature consistency across different identities for visually similar parts (e.g., similar poses or facial features). This ensures that the model captures structural commonalities while still allowing for identity-specific features at the global level.
>
>     2\. $L_{triplet}​$**​ and $L_{ce}​$.** On the other hand, $L_{triplet}​$​ and $L_{ce}​$​ are designed to maximize separability between identities. Triplet loss ensures that the model brings closer the features of the same identity while pushing apart features of different identities. Cross-entropy loss enforces separability at the identity level during classification.
>
>     3\. **No Intrinsic Conflict**. These losses work in tandem rather than conflict. The CILL loss operates at the local feature level, encouraging consistent local representations across identities, while $L_{triplet}​$​ and $L_{ce}​$ enforce global separability at the identity level. This means that while CILL helps the model recognize shared parts between identities,&#x20;
>
>     $L_{triplet}​$ and $L_{ce}​$ ensure that the model can still distinguish between different identities globally.
>
>     4\. **Synergistic Effect.** The synergy between CILL and the $L_{triplet}​$**​** and $L_{ce}​$**​** losses allows the model to learn both local consistency and global separability. This combination ensures that the model generalizes better to domain shifts while maintaining the ability to distinguish between different individuals effectively.

---

> > ### Author Response · Authors · 2025-11-21
> > **Response to Question 4 proposed by by Reviewer KjPj**
> >
> > *   **Question 4:**&#x20;
> >
> >     How much additional training overhead does ADSL introduce compared to the baseline model? Specifically the dual-stream forward pass, the memory bank, and the $L_{cos}​$ calculation.
> >
> >     **Reply to question 4:**&#x20;
> >
> >     Thanks for the reviewer's insightful question regarding the training overhead introduced by ADSL compared to the baseline model. Based on our experimental results, as shown in the Table 4 of the revised manuscript, we observed that the introduction of CILL and CILL + DAPS introduces additional training time per epoch, primarily due to the extra computational cost introduced by the dual-stream forward pass, memory bank updates, and the local cosine similarity loss $L_{cos}​$ calculation.
> >
> >     *   Baseline: The baseline model (without CILL or DAPS) has the lowest training time per epoch, at approximately 5.61 minutes for Market→Duke and 4.63 minutes for Duke→Market. This provides the benchmark for training overhead.
> >     *   Baseline + CILL: Introducing the CILL module increases the training time slightly to 5.72 minutes per epoch on Market→Duke and 4.74 minutes on Duke→Market, mainly due to the local feature consistency learning introduced by CILL.
> >     *   Baseline + CILL + DAPS: The addition of the Dual-Stream Adversarial Perturbation Strategy (DAPS) significantly increases the training time per epoch, with 16.2 minutes for Market→Duke and 13.5 minutes for Duke→Market. The additional overhead is due to the generation of adversarial perturbations for both clean and adversarial streams, as well as the memory bank updates required for handling these perturbations.
> >         Overall, the inference time remains constant across all configurations, with an average of 63.7 ms/img for Market→Duke and 30.2 ms/img for Duke→Market.
> >
> >     The ADSL framework introduces additional training time primarily due to the added complexity of the CILL module and the Dual-Stream Adversarial Perturbation Strategy (DAPS). While CILL slightly increases the training time, the addition of DAPS results in a more significant overhead due to the adversarial perturbation generation and memory bank updates. Despite these increases in training time, the inference time remains consistent across all configurations.

---

### Meta-Review · Area_Chair_x4Rg · 2026-01-21

**Summary:**

The paper has been reviewed by four reviewers and received initial 8, 6, 4 and 2 scores. The model tackles domain-generalized Person Re-ID by mining stable local commonalities and modeling local perturbations.

On balance, reviewers have had a number of concerns including the poor support between the importance of CILL and the ablation study, unbalanced contribution of two strategies, concerns about pose variations, concerns about the novelty and hyperparameters, effect of parts stability, among others.

On balance, meta-reviewer agrees with these shortcomings. Unfortunately, rebuttals did not manage to clarify many of these concerns.

Minor additional comments:
- Figure 1 is not self-explanatory - various encircled features are not explained.
- Related works on DG Re-ID mostly finishes on 2023 with exception of one cited paper from 2024
- Five loss components which highlights the complexity and engineered nature of the model

**Reviewer Concerns:**

Authors attempted to convince reviewers that parts stability is not relevant. However, concerns about the limited novelty, outdated related works, highly engineered pipeline with many loss components. The two components seem to have imbalanced contribution.

**Reviewer Scores:**

Reviewer 3tAx would likely decrease score below 6 due to lack of clear ablations regarding the effect of partitioning.

Reviewer KjPj would be likely still unconvinced about the importance of some of the components given their unbalanced nature.

Reviewer U4X6 would remain unconvinced about the novelty of models and would remain unconvinced about novelty of their combination given the large number of methods in the literature that rely on these components.

Reviewer uaiC would also still question the imbalance and likely maintain score below 6.

---

### Decision · Program_Chairs · 2026-01-26

Reject